# ACTION AND PERCEPTION AS DIVERGENCE MINIMIZATION

## ABSTRACT

We introduce a unified objective for action and perception of intelligent agents. Extending representation learning and control, we minimize the joint divergence between the combined system of agent and environment and a target distribution. Intuitively, such agents use perception to align their beliefs with the world, and use actions to align the world with their beliefs. Minimizing the joint divergence to an expressive target maximizes the mutual information between the agent's representations and inputs, thus inferring representations that are informative of past inputs and exploring future inputs that are informative of the representations. This lets us explain intrinsic objectives, such as representation learning, information gain, empowerment, and skill discovery from minimal assumptions. Moreover, interpreting the target distribution as a latent variable model suggests powerful world models as a path toward highly adaptive agents that seek large niches in their environments, rendering task rewards optional. The framework provides a common language for comparing a wide range of objectives, advances the understanding of latent variables for decision making, and offers a recipe for designing novel objectives. We recommend deriving future agent objectives the joint divergence to facilitate comparison, to point out the agent's target distribution, and to identify the intrinsic objective terms needed to reach that distribution.

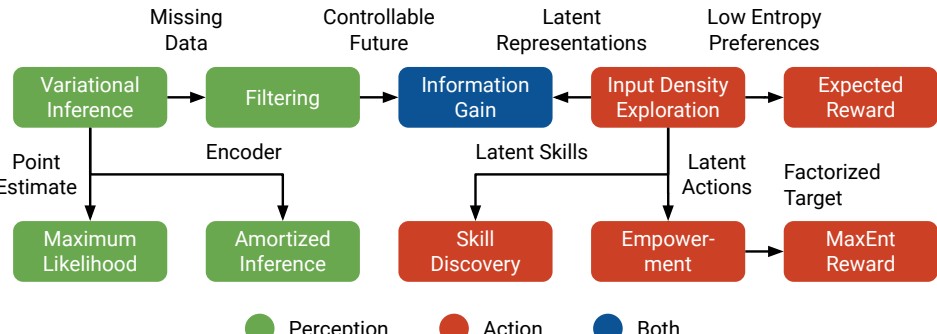

Figure 1: Overview of methods connected by the introduced framework of action and perception as divergence minimization. Each latent variable leads to a mutual information term between said variable and the data. The mutual information with past inputs explains representation learning. The mutual information with future inputs explains information gain, empowerment, and skill discovery. By leveraging multiple latent variables for the decision making process, agents can naturally combine multiple of the objectives. This figure shows the methods that drive from the well-established KL divergence and analogous method trees can be derived by choosing different divergence measures.

# 1 INTRODUCTION

To achieve goals in complex environments, intelligent agents need to perceive their environments and choose effective actions. These two processes, perception and action, are often studied in isolation. Despite the many objectives that have been proposed in the fields of representation learning and reinforcement learning, it remains unclear how the objectives relate to each other and which fundamentally new objectives remain yet to be discovered. Based on the KL divergence (Kullback and Leibler, 1951), we propose a unified framework for action and perception that connects a wide range of objectives to facilitate our understanding of them while providing a recipe for designing novel agent objectives. Our findings are conceptual in nature and this paper includes no empirical study. Instead, we offer a unified picture of a wide range of methods that have been shown to be successful in practice in prior work. The contributions of this paper are described as follows.

**Unified objective function for perception and action**    We propose joint KL minimization as a principled framework for designing and comparing agent objectives. KL minimization was proposed separately for perception as variational inference (Jordan et al., 1999; Alemi and Fischer, 2018) and for actions as KL control (Todorov, 2008; Kappen et al., 2009). Based on this insight, we formulate action and perception as jointly minimizing the KL from the world to a unified target distribution. The target serves both as the model to infer representations and as reward for actions. This extends variational inference to controllable inputs, while extending KL control to latent representations. We show a novel decomposition of joint KL divergence that explains several representation learning and exploration objectives. Divergence minimization additionally connects deep reinforcement learning to the free energy principle (Friston, 2010; 2019), while simplifying and overcoming limitations of its active inference implementations (Friston et al., 2017) that we discuss in Appendix B.

**Understanding latent variables for decision making**    Divergence minimization with an expressive target maximizes the mutual information between inputs and latents. Agents thus infer representations that are informative of past inputs and explore future inputs that are informative of the representations. For the past, this yields reconstruction (Hinton et al., 2006; Kingma and Welling, 2013) or contrastive learning (Gutmann and Hyvärinen, 2010; Oord et al., 2018). For the future, it yields information gain exploration (Lindley et al., 1956). Stochastic skills and actions are realized over time, so their past terms are constant. For the future, they lead to empowerment (Klyubin et al., 2005) and skill discovery (Gregor et al., 2016). RL as inference (Rawlik et al., 2010) does not maximize mutual information because its target is factorized. To optimize a consistent objective across past and future, latent representations should be accompanied by information gain exploration.

**Expressive world models for large ecological niches**    The more flexible an agent's target or model, the better the agent can adapt to its environment. Minimizing the divergence between the world and the model, the agent converges to a natural equilibrium or niche where it can accurately predict its inputs and that it can inhabit despite external perturbations (Schrödinger, 1944; Wiener, 1948; Haken, 1981; Friston, 2013; Berseth et al., 2019). While surprise minimization can lead to trivial solutions, divergence minimization encourages the niche to match the agent's model class, thus visiting all inputs proportionally to how well they can be understood. This suggests designing expressive world models of sensory inputs (Ebert et al., 2017; Hafner et al., 2018; Gregor et al., 2019) as a path toward building highly adaptive agents, while rendering task rewards optional.

# 2 FRAMEWORK

This section introduces the framework of action and perception as divergence minimization (APD). To unify action and perception, we formulate the two processes as joint KL minimization with a shared target distribution. The target distribution expresses the agent's preferences over system configurations and is also the probabilistic model under which the agent infers its representations. Using an expressive model as the target maximizes the mutual information between the latent variables and the sequence of sensory inputs, thus inferring latent representations that are informative of past inputs and exploring future inputs that are informative of the representations. We assume knowledge of basic concepts from probability and information theory that are reviewed in Appendix D.

## 2.1 JOINT KL MINIMIZATION

Consider a stochastic system described by a joint probability distribution over random variables. For example, the random variables for supervised learning are the inputs and labels and for an agent they are the sequence of sensory inputs, internal representations, and actions. More generally, we combine

| Formulation | Preferences | Latent Entropy | Input Entropy |
|---|---|---|---|
| Divergence Minimization | ✓ | ✓ | ✓ |
| Active Inference | ✓ | ✓ | ✗ |
| Expected Reward | ✓ | ✗ | ✗ |

Table 1: High-level comparison of different agent objectives. All objectives express preferences over system configurations as a scalar value. Active inference additionally encourages entropic latents. Divergence minimization additionally encourages entropic inputs. Active inference makes additional choices about the optimization, as detailed in Appendix B, and the motivation for our work is in part to offer a simpler alternative to active inference. We show that when using expressive models as preferences, the entropy terms result in a wide range of task-agnostic agent objectives.

all input variables into $x$ and the remaining variables that we term latents into $z$. We will see that different latents correspond to different representation learning and exploration objectives.

The random variables are distributed according to their generative process or actual distribution $p_\phi$. Parts of the actual distribution can be unknown, such as the data distribution, and parts can be influenced by varying the parameter vector $\phi$, such as the distribution of stochastic representations or actions. As a counterpart to the actual distribution, we define the desired target distribution $\tau$ over the same support. It describes our preferences over system configurations and can be unnormalized,

$$\text{Actual distribution:} \quad x, z \sim p_\phi(x, z) \qquad \text{Target distribution:} \quad \tau(x, z). \qquad (1)$$

We formulate the problem of joint KL minimization as changing the parameters $\phi$ to bring the actual distribution of all random variables as close as possible to the target distribution, as measured by the KL divergence (Kullback and Leibler, 1951; Li et al., 2017; Alemi and Fischer, 2018),

$$\min_\phi \text{KL}\big[p_\phi(x, z) \,\big\|\, \tau(x, z)\big]. \qquad (2)$$

All expectations and KLs throughout the paper are integrals under the actual distribution, so they can be estimated from samples of the system and depend on $\phi$. Equation 2 is the reverse KL or information projection used in variational inference (Csiszár and Matus, 2003).

**Examples**   For representation learning, $p_\phi$ is the joint of data and belief distributions and $\tau$ is a latent variable model. Note that we use $p_\phi$ to denote not the model under which we infer beliefs but the generative process of inputs and their representations. For control, $p_\phi$ is the trajectory distribution under the current policy and $\tau$ corresponds to the utility of the trajectory. The parameters $\phi$ include everything the optimizer can change directly, such as sufficient statistics of representations, model parameters, and policy parameters.

**Target parameters**   There are two ways to denote deterministic values within our framework, also known as MAP estimates in the probabilistic modeling literature (Bishop, 2006). We can either use a fixed target distribution and use a latent variable that follows a point mass distribution (Dirac, 1958), or we explicitly parameterize the target using a deterministic parameter as $\tau_\phi$. In either case, $\tau$ refers to the fixed model class. The two approaches are equivalent because in both cases the target receives a deterministic value that has no entropy regularizer. For more details, see Appendix A.1.

**Assumptions**   Divergence minimization uses only two inductive biases, namely that the agent optimizes an objective and that it uses random variables to represent uncertainty. Choosing the well-established KL as the divergence measure is an additional assumption. It corresponds to maximizing the expected log probability under the target while encouraging high entropy for all variables in the system to avoid overconfidence, as detailed in Appendix C. Common objectives with different degrees of entropy regularization are summarized in Table 1.

**Generality**   Alternative divergence measures would lead to different optimization dynamics, different solutions if the target cannot be reached, and potentially novel objectives for representation learning and exploration. Nonetheless, the KL can describe any converged system, trivially by choosing its actual distribution as the target, and thus offers a simple and complete mathematical perspective for comparing a wide range of specific objectives that correspond to different latent variables and target distributions.

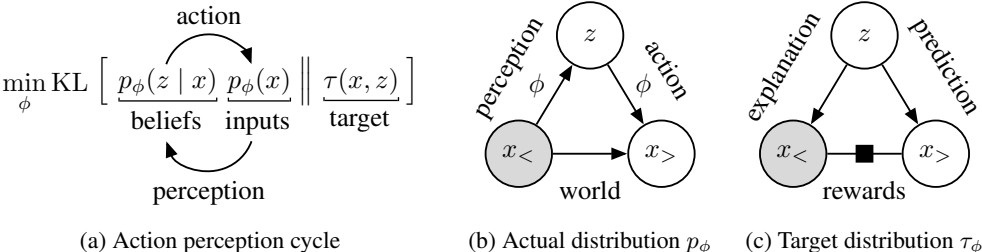

(a) Action perception cycle     (b) Actual distribution $p_\phi$     (c) Target distribution $\tau_\phi$

Figure 2: Action and perception minimize the joint KL divergence to a unified target distribution that can be interpreted as a learning probabilistic model of the system. Given the target, perception aligns the agent's beliefs with past inputs while actions align future inputs with its beliefs. There are many ways to specify the target, for example as a latent variable model that explains past inputs and predicts future inputs and an optional reward factor that is shown as a filled square.

## 2.2 INFORMATION BOUNDS

We show that for expressive targets that capture dependencies between the variables in the system, minimizing the joint KL increases both the preferences and the mutual information between inputs $x$ and latents $z$. This property allows divergence minimization to explain a wide range of existing representation learning and exploration objectives. We use the term representation learning for inferring deterministic or stochastic variables from inputs, which includes local representations of individual inputs and global representations such as model parameters.

**Latent preferences**    The joint KL can be decomposed in multiple ways, for example into a marginal KL plus a conditional KL or by grouping marginal with conditional terms. To reveal the mutual information maximization, we decompose the joint KL into a preference seeking term and an information seeking term. The decomposition can be done either with the information term expressed over inputs and the preferences expressed over latents or the other way around,

$$\underbrace{\mathrm{KL}\big[p_\phi(x,z) \,\big\|\, \tau(x,z)\big]}_{\text{joint divergence}} = \underbrace{\mathrm{E}\,\mathrm{KL}\big[p_\phi(z \mid x) \,\big\|\, \tau(z)\big]}_{\text{realizing latent preferences}} - \underbrace{\mathrm{E}\big[\ln \tau(x \mid z) - \ln p_\phi(x)\big]}_{\text{information bound}}. \qquad (3)$$

All expectations throughout the paper are over all variables, under the actual distribution, and thus depend on the parameters $\phi$. The first term on the right side of Equation 3 is a KL regularizer that keeps the belief $p_\phi(z \mid x)$ over latent variables close to the marginal latent preferences $\tau(z)$. The second term is a variational bound on the mutual information $\mathrm{I}\big[x; z\big]$ (Barber and Agakov, 2003). The bound is expressed in input space. Maximizing the conditional $\ln \tau(x \mid z)$ seeks latent variables that accurately predict inputs while minimizing the marginal $\ln p_\phi(x)$ seeks diverse inputs.

**Variational free energy**    When the agent cannot influence its inputs, such as when learning from a fixed dataset, the input entropy $\mathrm{E}\big[-\ln p_\phi(x)\big]$ is not parameterized and can be dropped from Equation 3. This yields the free energy or ELBO objective used by variational inference to infer approximate posterior beliefs in latent variable models (Hinton and Van Camp, 1993; Jordan et al., 1999). The free energy regularizes the belief $p_\phi(z \mid x)$ to stay close to the prior $\tau(z)$ while reconstructing inputs via $\tau(x \mid z)$. However, in reinforcement and active learning, inputs can be influenced and thus the input entropy should be kept, which makes the information bound explicit.

**Input preferences**    Analogously, we decompose the joint KL the other way around. The first term on the right side of Equation 4 is a KL regularizer that keeps the conditional input distribution $p_\phi(x \mid z)$ close to the marginal input preferences $\tau(x)$. This term is analogous to the objective in KL control (Todorov, 2008; Kappen et al., 2009), except that the inputs now depend upon latent variables via the policy. The second term is again a variational bound on the mutual information $\mathrm{I}\big[x; z\big]$, this time expressed in latent space. Intuitively, the bound compares the belief $\tau(z \mid x)$ after observing the inputs and the belief $p_\phi(z)$ before observing any inputs to measure the gained information,

$$\underbrace{\mathrm{KL}\big[p_\phi(x,z) \,\big\|\, \tau(x,z)\big]}_{\text{joint divergence}} = \underbrace{\mathrm{E}\,\mathrm{KL}\big[p_\phi(x \mid z) \,\big\|\, \tau(x)\big]}_{\text{realizing input preferences}} - \underbrace{\mathrm{E}\big[\ln \tau(z \mid x) - \ln p_\phi(z)\big]}_{\text{information bound}}. \qquad (4)$$

The information bounds are tighter the better the target conditional approximates the actual conditional, meaning that the agent becomes better at maximizing mutual information as it learns more about the relation between the two variables. This requires an expressive target that captures correlations between inputs and latents, such as a latent variable model or deep neural network. Maximizing the mutual information accounts for both learning latent representations that are informative of inputs as well as exploring inputs that are informative of the latent representations.

### 2.3 MODELS AS PREFERENCES

The target distribution defines our preferences over system configurations. However, we can also interpret it as a probabilistic model, or energy-based model if unnormalized (LeCun et al., 2006). This is because minimizing the joint KL infers beliefs over latent variables that approximate the posteriors under the model, as shown in Section 2.2. Because the target is not parameterized, it corresponds to the fixed model class, with parameters being inferred as latent variables, optionally using point mass distributions. As the agent brings the actual distribution closer to the target, the target also becomes a better predictor of the actual distribution. Divergence minimization thus emphasizes that the model class simply expresses preferences over latent representations and inputs and lets us interpret inference as bringing the joint of data and belief distributions toward the model joint.

**Input preferences** Minimizing the joint divergence also minimizes the divergence between the agent's input distribution $p_\phi(x)$ and the marginal input distribution under its target or model $\tau(x)$. The marginal input distribution of the model is thus the agent's preferred input distribution, that the agent aims to sample from in the environment. Because $\tau(x)$ marginalizes out all latent variables and parameters, it describes how well an input sequence $x$ can possibly be described by the model class, as used in the Bayes factor (Jeffreys, 1935; Kass and Raftery, 1995). Divergence minimizing agents thus seek out inputs proportionally to how their models can learn to predict them through inference, while avoiding inputs that are inherently unpredictable given their model class. Because the target can be unnormalized, we can combine a latent variable model with a reward factor of the form $\exp(r(x))$ to create a target that incorporates task rewards. The reward factor adds preferences for certain inputs without affecting the remaining variables in the model. We describe examples of such reward factors this in Appendix A.4 and Section 3.1.

**Action perception cycle** Interpreting the target as a model shows that divergence minimization is consistent with the idea of perception as inference suggested by Helmholtz (Helmholtz, 1866; Gregory, 1980). Expressing preferences as models is inspired by the free energy principle and active inference (Friston, 2010; Friston et al., 2012; 2017), which we compare to in Appendix B. Divergence minimization inherits an interpretation of action and perception from active inference that we visualize in Figure 2a. While action and perception both minimize the same joint KL, they affect different variables. Perception is based on inputs and affects the beliefs over representations, while actions are based on the representations and affect inputs. Given a unified target, perception thus aligns the agent's beliefs with the world while actions align the world with its beliefs.

**Niche seeking** The information bounds responsible for representation learning and exploration are tighter under expressive targets, as shown in Section 2.2. What happens when we move beyond task rewards and simply define the target as a flexible model? The more flexible the target and belief family, the better the agent can minimize the joint KL. Eventually, the agent will converge to a natural equilibrium or ecological niche where it can predict its inputs well and that it can inhabit despite external perturbations (Wiener, 1948; Ashby, 1961). Niche seeking connects to surprise minimization (Schrödinger, 1944; Friston, 2013; Berseth et al., 2019), which aims to *maximize* the marginal likelihood of inputs under a model. In environments without external perturbations, this can lead to trivial solutions once they are explored. Divergence minimization instead aims to *match* the marginal input distribution of the model. This encourages large niches that cover all inputs that the agent can learn to predict. Moreover, it suggests that expressive world models lead to autonomous agents that understand and inhabit large niches, rendering task rewards optional.

### 2.4 PAST AND FUTURE

Representations are computed from past inputs and exploration targets future inputs. To identify the two processes, we thus need to consider how an agent optimizes the joint KL after observing past inputs $x_<$ and before observing future inputs $x_>$, as discussed in Figure 2b. For example, past inputs can be stored in an experience dataset and future inputs can be approximated by planning with a learned world model, on-policy trajectories, or replay of past inputs (Sutton, 1991). To condition the joint KL on past inputs, we first split the information bound in Equation 3 into two smaller bounds on the past mutual information $\mathrm{I}[x_<; z]$ and additional future mutual information $\mathrm{I}[x_>; z \mid x_<]$,

$$\underbrace{\mathrm{E}\big[\ln \tau(z \mid x) - \ln p_\phi(z)\big]}_{\text{information bound}} = \mathrm{E}\big[\ln \tau(z \mid x) - \ln p_\phi(z \mid x_<) + \ln p_\phi(z \mid x_<) - \ln p_\phi(z)\big]$$
$$\geq \mathrm{E}\big[\underbrace{\ln \tau(z \mid x) - \ln p_\phi(z \mid x_<)}_{\text{future information bound}} + \underbrace{\ln \tau(z \mid x_<) - \ln p_\phi(z)}_{\text{past information bound}}\big]. \quad (5)$$

| Latent | Target | Past Term | Future Term | Agents |
|--------|--------|-----------|-------------|--------|
| Actions | Factorized | — | Action entropy | A3C, SQL, SAC |
| Actions | Expressive | — | Empowerment | VIM, ACIE, EPC |
| Skills | Expressive | — | Skill discovery | VIC, SNN, DIAYN, VALOR |
| States | Expressive | Repr. learning | Information gain | NDIGO, DVBF-LM |
| Parameters | Expressive | Model learning | Information gain | VIME, MAX, Plan2Explore |

Table 2: Divergence minimization accounts for a wide range of agent objectives. Each latent variable used by the agent contributes a future objective term. Moreover, latent variables that are not observed over time, such as latent representations and model parameters, additionally each contribute a past objective term. Combining multiple latent variables combines their objective terms. Refer to Section 3 for detailed derivations of these individual examples and citations of the listed agents.

Equation 5 splits the belief update from the prior $p_\phi(z)$ to the posterior $\tau(z \mid x)$ into two updates via the intermediate belief $p_\phi(z \mid x_<)$ and then applies the variational bound from Barber and Agakov (2003) to allow both updates to be approximate. Splitting the information bound lets us separate past and future terms in the joint KL, or even separate individual time steps. It also lets us separately choose to express terms in input or latent space. This decomposition is one of our main contributions and shows how the joint KL divergence accounts for both representation learning and exploration,

$$
\begin{aligned}
\mathrm{KL}\big[p_\phi(x,z) \,\big\|\, \tau(x,z)\big] \leq &\underbrace{\mathrm{E\,KL}\big[p_\phi(z \mid x_<) \,\big\|\, \tau(z)\big]}_{\text{realizing past latent preferences}} - \underbrace{\mathrm{E}\big[\ln \tau(x_< \mid z) - \ln p_\phi(x_<)\big]}_{\text{representation learning}} \\
&+ \underbrace{\mathrm{E\,KL}\big[p_\phi(x_> \mid x_<,z) \,\big\|\, \tau(x_> \mid x_<)\big]}_{\text{realizing future input preferences}} - \underbrace{\mathrm{E}\big[\ln \tau(z \mid x) - \ln p_\phi(z \mid x_<)\big]}_{\text{exploration}}.
\end{aligned}
\tag{6}
$$

Conditioning on past inputs $x_<$ removes their expectation and renders $p_\phi(x_<)$ constant. While some latent variables in the set $z$ are never realized, such as latent state estimates or model parameters, other latent variables become observed over time, such as stochastic actions or skills. Because the agent selects the values of these variables, we have to condition the objective terms on them as causal interventions (Pearl, 1995; Ortega and Braun, 2010). In practice, this means replacing all occurrences of $z$ by the unobserved latents $z_>$ and conditioning those terms on the observed latents $\mathrm{do}(z_<)$. To keep notation simple, we omit this step in our notation.

To build an intuition about Equation 6, we discuss the four terms on the right-hand side. The first two terms involve the past while the last two terms involve the future. The first term keeps the agent's belief $p_\phi(z \mid x_<)$ close to the prior $\tau(z)$ to incorporate inductive biases. The second term encourages the belief to be informative of the past inputs so that the inputs are reconstructed by $\tau(x_< \mid z)$, where is $p_\phi(x_<)$ is a constant because $x_<$ are observed. The third term is the control objective that steers toward future inputs that match the preferred input distribution $\tau(x_> \mid x_<)$. The fourth term is an information bound that seeks out future inputs that are informative of the latent representations in $z$ and encourages actions or skills in $z$ that maximally influence future inputs.

The decomposition shows that the joint KL accounts for both learning informative representations of past inputs and exploring informative future inputs as two sides of the same coin. From this, we derive several representation and exploration objectives by including different latent variables in the set $z$. These objectives are summarized in Table 2 and derived with detailed examples in Section 3.

**Representation learning**   Because past inputs are observed, the past information bound only affects the latents. Expressed as Equation 3, it leads to reconstruction (Hinton et al., 2006), and as Equation 4, it leads to contrastive learning (Gutmann and Hyvärinen, 2010; Oord et al., 2018). This accounts for local representations of individual inputs, as well as global representations, such as latent parameters. Moreover, representations can be inferred online or amortized using an encoder (Kingma and Welling, 2013). Latents with point estimates are equivalent to target parameters and thus are optimized jointly to tighten the variational bounds. Because past actions and skills are realized, their mutual information with realized past inputs is constant and thus contributes no past objective terms.

**Exploration**   Under a flexible target, latents in $z$ result in information-maximizing exploration. For latent representations, this is known as expected information gain and encourages informative future inputs that convey the most bits about the latent variable, such as world model parameters, policy parameters, or state estimates (Lindley et al., 1956; Sun et al., 2011). For stochastic actions, a fully factorized target leads to maximum entropy RL. An expressive target yields empowerment,

maximizing the agent's influence over the world (Klyubin et al., 2005). For skills, it yields skill discovery or diversity that learns distinct modes of behavior that together cover many different trajectories (Gregor et al., 2016; Florensa et al., 2017; Eysenbach et al., 2018; Achiam et al., 2018).

## 3    EXAMPLES

We use the framework of action and perception as divergence minimization presented in Section 2 to derive a wide range of concrete objective functions that have been proposed in the literature, shown in Figure 1. For this, we analyze the cases of different latent variables and factorizations of the actual and target distributions. These derivations serve as practical examples for producing new objective functions within our framework. We start by describing maximum entropy RL because of its popularity in the literature. Due to space constraints, we refer to Appendix A for the remaining examples, which include variational inference, amortized inference, filtering, KL control, empowerment, skill discovery, and information gain.

**Designing novel objectives**    In practice, an agent is determined by its target distribution, belief family, and optimization algorithm. Our framework thus suggests to break down the implementation of an agent into the same three components that are typically considered in probabilistic modeling. As Section 2 showed, the target distribution is also the model under which the agent infers its beliefs about the world. We also saw that more expressive models allow agents to increase the mutual information between their inputs and latents more. To design an agent that learns a lot about the world, we should thus design expressive world model and use them as the target distribution. For example, these could include latent state estimates, latent parameters, latent skills, hierarchical latents, or temporal abstraction. Each world model corresponds to a new agent objective.

### 3.1    MAXIMUM ENTROPY RL

Maximum entropy RL (Williams and Peng, 1991; Kappen et al., 2009; Rawlik et al., 2010; Tishby and Polani, 2011; Fox et al., 2015; Schulman et al., 2017; Haarnoja et al., 2018) chooses stochastic actions to maximize a task reward while remaining close to an action prior. The action prior is typically independent of the inputs, corresponding to a factorized target. The objective thus does not contain a mutual information term. Despite factorized targets being common in practice, we suggest that expressive targets, such as world models, are preferable in the longer term.

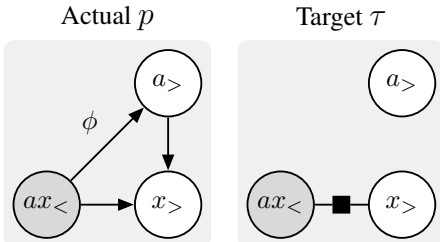

Figure 3: Maximum Entropy RL

Figure 3 shows the actual and target distributions for maximum entropy RL. The input sequence is $x \doteq \{x_t\}$ and the action sequence is $a \doteq \{a_t\}$. In the graphical model, these are grouped into past actions and inputs $ax_<$, future actions $a_>$, and future inputs $x_>$. The actual distribution consists of the fixed environment dynamics and a stochastic policy. The target consists of a reward factor, an action prior that is often the same for all time steps, and the environment dynamics,

$$\text{Actual:} \quad p_\phi(x, a) \doteq \prod_t \underbrace{p(x_t \mid x_{1:t-1}, a_{1:t-1})}_{\text{environment}} \underbrace{p_\phi(a_t \mid x_{1:t}, a_{1:t-1})}_{\text{policy}},$$

$$\text{Target:} \quad \tau(x, a) \dot\propto \prod_t \exp(\underbrace{r(x_t)}_{\text{reward}}) \underbrace{p(x_t \mid x_{1:t-1}, a_{1:t-1})}_{\text{environment}} \underbrace{\tau(a_t)}_{\text{action prior}}. \tag{7}$$

Minimizing the joint KL results in a complexity regularizer in action space and the expected reward. Including the environment dynamics in the target cancels out the curiosity term as in the expected reward case in Appendix A.4, leaving maximum entropy RL to explore only in action space. Moreover, including the environment dynamics in the target gives up direct control over the agent's input preferences, as they depend not just on the reward but also the environment dynamics marginal. Because the target distribution is factorized and does not capture dependencies between $x$ and $a$, maximum entropy RL does not maximize their mutual information,

$$\text{KL}\big[p_\phi \,\big\|\, \tau\big] = \sum_t \underbrace{\text{E}\,\text{KL}\big[p_\phi(a_t \mid x_{1:t}, a_{1:t-1}) \,\big\|\, \tau(a_t)\big]}_{\text{complexity}} - \underbrace{\text{E}\big[r(x_t)\big]}_{\text{expected reward}}. \tag{8}$$

The action complexity KL can be simplified into an entropy regularizer by choosing a uniform action prior as in SQL (Haarnoja et al., 2017) and SAC (Haarnoja et al., 2018). The action prior

can also depend on the past inputs and incorporate knowledge from previous tasks as in Distral (Teh et al., 2017) and work by Tirumala et al. (2019) and Galashov et al. (2019). Divergence minimization motivates combining maximum entropy RL with input density exploration by removing the environment dynamics from the target distribution. The resulting agent aims to converge to the input distribution that is proportional to the exponentiated task reward. Moreover, divergence minimization shows that the difference between maximum entropy RL and empowerment, that we describe in Appendix A.5, is the target factorizes actions and inputs or captures their dependencies.

## 4 RELATED WORK

**Divergence minimization** Various problems have been formulated as minimizing a divergence between two distributions. TherML (Alemi and Fischer, 2018) studies representation learning as KL minimization. We follow their interpretation of the data and belief as actual distribution, although their target is only defined by its factorization. ALICE (Li et al., 2017) describes adversarial learning as joint distribution matching, while Kirsch et al. (2020) unify information-based objectives. Ghasemipour et al. (2019) describe imitation learning as minimizing divergences between the inputs of learned and expert behavior. None of these works consider combined representation learning and control. Thompson sampling minimizes the forward KL to explain action and perception as exact inference (Ortega and Braun, 2010). In comparison, we optimize the backward KL to support intractable models and connect to a wide range of practical objectives.

**Active inference** The presented framework is inspired by the free energy principle, which studies the dynamics of agent and environment as stationary SDEs (Friston, 2010; 2019). We inherit the interpretations of active inference, which implements agents based on the free energy principle (Friston et al., 2017). While divergence minimization matches the input distribution under the model, active inference maximizes the probability of inputs under it, resulting in smaller niches. Moreover, active inference optimizes the exploration terms only with respect to actions, which requires a specific action prior. Finally, typical implementations of active inference involve an expensive Bayesian model average over possible action sequences, limiting its applications to date (Friston et al., 2015; 2020). We compare to active inference in detail in Appendix B. Generalized free energy (Parr and Friston, 2019) studies a unified objective similar to ours, although its entropy terms are defined heuristically rather than derived from a general principle.

**Control as inference** It is well known that RL can be formulated as KL minimization over inputs and actions (Todorov, 2008; Kappen et al., 2009; Rawlik et al., 2010; Ortega and Braun, 2011; Levine, 2018), as well as skills (Hausman et al., 2018; Tirumala et al., 2019; Galashov et al., 2019). We build upon this literature and extend it to agents with latent representations, leading to variational inference on past inputs and information seeking exploration for future inputs. Divergence minimization relates the above methods and motivates an additional entropy regularizer for inputs (Todorov, 2008; Lee et al., 2019b; Xin et al., 2020). SLAC (Lee et al., 2019a) combines representation learning and control but does not consider the future mutual information, so their objective changes over time. In comparison, we derive the terms from a general principle and point out the information gain that results in an objective that is consistent over time. The information gain term may also address concerns about maximum entropy RL raised by O'Donoghue et al. (2020).

## 5 CONCLUSION

We introduce a general objective for action and perception of intelligent agents, based on minimizing the KL divergence. To unify the two processes, we formulate them as joint KL minimization with a shared target distribution. This target distribution is the probabilistic model under which the agent infers its representations and expresses the agent's preferences over system configurations. We summarize the key takeaways as follows:

- **Unified objective for action and perception** Divergence minimization with an expressive target maximizes the mutual information between latents and inputs. This leads to inferring representations that are informative of past inputs and exploration of future inputs that are informative of the representations. To optimize a consistent objective that does not change over time, any latent representation should be accompanied by a corresponding exploration term.

- **Understanding of latent variables for decision making** Different latents lead to different objective terms. Latent representations are never observed, leading to both representation learning

and information gain exploration. Actions and skills become observed over time and thus do not encourage representation learning but lead to generalized empowerment and skill discovery.

- **Adaptive agents through expressive world models**    Divergence minimization agents with an expressive target find niches where they can accurately predict their inputs and that they can inhabit despite external perturbations. The niches correspond to the inputs that the agent can learn to understand, which is facilitated by the exploration terms. This suggests designing powerful world models as a path toward building autonomous agents, without the need for task rewards.

- **General recipe for designing novel objectives**    When introducing new agent objectives, we recommend deriving them from the joint KL by choosing a latent structure and target. For information maximizing agents, the target is an expressive model, leaving different latent structures to be explored. Deriving novel objectives from the joint KL facilitates comparison, renders explicit the target distribution, and highlights the intrinsic objective terms needed to reach that distribution.

- **Discovering new families of agent objectives**    Our work shows that a family of representation learning and exploration objectives can be derived from minimizing a joint KL between the system and a target distribution. Different divergence measures give rise to new families of such agent objectives that could be easier to optimize or converge to better optima for infeasible targets. We leave exploring those objective families and comparing them empirically as future work.

Without constraining the class of targets, our framework is general and can describe any system. This by itself offers a framework for comparing many existing methods. However, interpreting the target as a model further suggests that intelligent agents may use especially expressive models as targets. This hypothesis should be investigated in future work by examining artificial agents with expressive world models or by modeling the behavior of natural agents as divergence minimization.

**Acknowledgements**    Hidden for review.

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

## A ADDITIONAL EXAMPLES

This section leverages the presented framework to explain a wide range of objectives in a unifying review, as outlined in Figure 1. For this, we include different variables in the actual distribution, choose different target distributions, and then rewrite the joint KL to recover familiar objectives. We start with perception, the case with latent representations but uncontrollable inputs and then turn to action, the case without latent representations but with controllable inputs. We then turn to combined action and perception. The derivations follow the general recipe described in Section 2. The same steps can be followed for new latent structures and target distributions to yield novel agent objectives.

### A.1 VARIATIONAL INFERENCE

Following Helmholtz, we describe perception as inference under a model (Helmholtz, 1866; Gregory, 1980; Dayan et al., 1995). Inference computes a posterior over representations by conditioning the model on inputs. Because this has no closed form in general, variational inference optimizes a parameterized belief to approximate the posterior (Peterson, 1987; Hinton and Van Camp, 1993; Jordan et al., 1999).

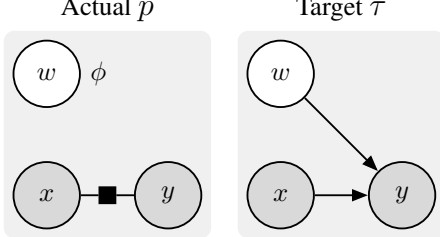

Figure 4: Variational Inference

Figure 4 shows variational inference for the example of supervised learning using a BNN (Denker et al., 1987; MacKay, 1992a; Blundell et al., 2015). The inputs are images $x \doteq \{x_i\}$ and their classes $y \doteq \{y_i\}$ and we infer the latent parameters $w$ as a global representation of the data set (Alemi and Fischer, 2018). The parameters depend on the inputs only through the optimization process that produces $\phi$. The target consists of a parameter prior and a conditional likelihood that uses the parameters to predict classes from images,

$$
\begin{aligned}
\text{Actual:} \quad & p_\phi(x, y, w) \doteq \underbrace{p_\phi(w)}_{\text{belief}} \textstyle\prod_i \underbrace{p(x_i, y_i)}_{\text{data}}, \\
\text{Target:} \quad & \tau(x, y, w) \stackrel{.}{\propto} \underbrace{\tau(w)}_{\text{prior}} \textstyle\prod_i \underbrace{\tau(y_i \mid x_i, w)}_{\text{likelihood}}.
\end{aligned} \tag{9}
$$

Applying the framework, we minimize the KL between the actual and target joints. Because the data distribution is fixed here, the input marginal $p(x, y)$ is a constant. In this case, the KL famously results in the free energy or ELBO objective (Hinton and Van Camp, 1993; Jordan et al., 1999) that trades off remaining close to the prior and enabling accurate predictions. The objective can be interpreted as the description length of the data set under entropy coding (Huffman, 1952; MacKay, 2003) because it measures the nats needed for storing both parameter belief and prediction residuals,

$$
\mathrm{KL}\big[p_\phi \,\big\|\, \tau\big] = \underbrace{\mathrm{KL}\big[p_\phi(w) \,\big\|\, \tau(w)\big]}_{\text{complexity}} - \underbrace{\mathrm{E}\big[\ln \tau(y \mid x, w)\big]}_{\text{accuracy}} + \underbrace{\mathrm{E}\big[\ln p(x, y)\big]}_{\text{constant}}. \tag{10}
$$

Variational methods for BNNs (Peterson, 1987; Hinton and Van Camp, 1993; Blundell et al., 2015) differ in their choices of prior and belief distributions and inference algorithm. This includes hierarchical priors (Louizos and Welling, 2016; Ghosh and Doshi-Velez, 2017), data priors (Louizos and Welling, 2016; Hafner et al., 2019b; Sun et al., 2019), flexible posteriors (Louizos and Welling, 2016; Sun et al., 2017; Louizos and Welling, 2017; Zhang et al., 2018; Chang et al., 2019), low rank posteriors (Izmailov et al., 2018; Dusenberry et al., 2020), and improved inference algorithms (Wen et al., 2018; Immer et al., 2020). BNNs have been leveraged for RL for robustness (Okada et al., 2020; Tran et al., 2019) and exploration (Houthooft et al., 2016; Azizzadenesheli et al., 2018).

**Target parameters**  While expressive beliefs over model parameters lead to a global search for their values, provide uncertainty estimates for predictions, and enable directed exploration in the RL setting, they can be computationally expensive. When these properties are not needed, we can choose a point mass distribution $p_\phi(w) \to \delta_\phi(w) \doteq \{1 \text{ if } w = \phi \text{ else } 0\}$ to simplify the expectations and avoid the entropy and mutual information terms that are zero for this variable (Dirac, 1958),

$$
\underbrace{\mathrm{KL}\big[p_\phi(w) \,\big\|\, \tau(w)\big]}_{\text{complexity}} - \underbrace{\mathrm{E}\big[\ln \tau(y \mid x, w)\big]}_{\text{accuracy}} \to \underbrace{\ln \tau(\phi)}_{\text{complexity}} - \underbrace{\mathrm{E}\big[\ln \tau(y \mid x, \phi)\big]}_{\text{accuracy}} \doteq \underbrace{\mathrm{E}\big[-\ln \tau_\phi(y \mid x)\big]}_{\text{parameterized target}}. \tag{11}
$$

Point mass beliefs result in MAP or maximum likelihood estimates (Bishop, 2006; Murphy, 2012) that are equivalent to parameterizing the target as $\tau_\phi$. Parameterizing the target is thus a notational

choice for random variables with point mass beliefs. Technically, we also require the prior over target parameters to be integrable but this is true in practice where only finite parameter spaces exist.

## A.2 AMORTIZED INFERENCE

Local representations represent individual inputs. They can summarize inputs more compactly, enable interpolation between inputs, and facilitate generalization to unseen inputs. In this case, we can use amortized inference (Kingma and Welling, 2013; Rezende et al., 2014; Ha et al., 2016) to learn an encoder that maps each input to its corresponding belief. The encoder is shared among inputs to reuse computation. It can also compute beliefs for new inputs without further optimization, although optimization can refine the belief (Kim et al., 2018).

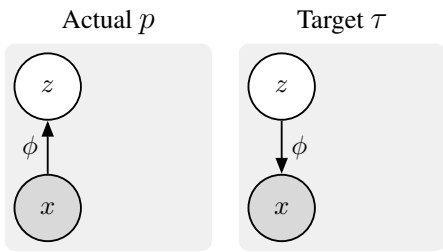

Figure 5: Amortized Inference

Figure 5 shows amortized inference on the example of a VAE (Kingma and Welling, 2013; Rezende et al., 2014). The inputs are images $x \doteq \{x_i\}$ and we infer their latent codes $z = \{z_i\}$. The actual distribution consists of the unknown and fixed data distribution and the parameterized encoder $p_\phi(z_i \mid x_i)$. The target is a probabilistic model defined as the prior over codes and the decoder that computes the conditional likelihood of each image given its code. We parameterize the target here, but one could also introduce an additional latent variable to infer a distribution over decoder parameters as in Appendix A.1,

$$
\begin{aligned}
\text{Actual:} \quad & p_\phi(x, z) \doteq \prod_i \underbrace{p(x_i)}_{\text{data}} \underbrace{p_\phi(z_i \mid x_i)}_{\text{encoder}}, \\
\text{Target:} \quad & \tau_\phi(x, z) \doteq \prod_i \underbrace{\tau_\phi(x_i \mid z_i)}_{\text{decoder}} \underbrace{\tau(z_i)}_{\text{prior}}.
\end{aligned}
\tag{12}
$$

Because the data distribution is still fixed, minimizing the joint KL again results in the variational free energy or ELBO objective that trades of prediction accuracy and belief simplicity. However, by including the constant input marginal, we highlight that the prediction term is a variational bound on the mutual information that encourages the representations to be informative of their inputs,

$$
\mathrm{KL}\big[p_\phi \,\big\|\, \tau_\phi\big] = \underbrace{\mathrm{E}\,\mathrm{KL}\big[p_\phi(z \mid x) \,\big\|\, \tau(z)\big]}_{\text{complexity}} - \underbrace{\mathrm{E}\big[\ln \tau_\phi(x \mid z) - \ln p(x)\big]}_{\text{information bound}}.
\tag{13}
$$

In input space, the information bound leads to reconstruction as in DBNs (Hinton et al., 2006), VAEs (Kingma and Welling, 2013; Rezende et al., 2014), and latent dynamics (Krishnan et al., 2015; Karl et al., 2016). In latent space, it leads to contrastive learning as in NCE (Gutmann and Hyvärinen, 2010), CPC (Oord et al., 2018; Guo et al., 2018), CEB (Fischer, 2020), and SimCLR (Chen et al., 2020). To maximize their mutual information, $x$ and $z$ should be strongly correlated under the target distribution, which explains the empirical benefits of ramping up the decoder variance throughout learning (Bowman et al., 2015; Eslami et al., 2018) or scaling the temperature of the contrastive loss (Chen et al., 2020). The target defines the variational family and includes inductive biases (Tschannen et al., 2019). Both forms have enabled learning world models for planning (Ebert et al., 2018; Ha and Schmidhuber, 2018; Zhang et al., 2019; Hafner et al., 2018; 2019a) and accelerated RL (Lange and Riedmiller, 2010; Jaderberg et al., 2016; Lee et al., 2019a; Yarats et al., 2019; Gregor et al., 2019).

## A.3 FUTURE INPUTS

Before moving to actions, we discuss perception with unobserved future inputs that are outside of our control (Ghahramani and Jordan, 1995). This is typical in supervised learning where the test set is unavailable during training (Bishop, 2006), in online learning where training inputs become available over time (Amari, 1967), and in filtering where only inputs up to the current time are available (Kalman, 1960).

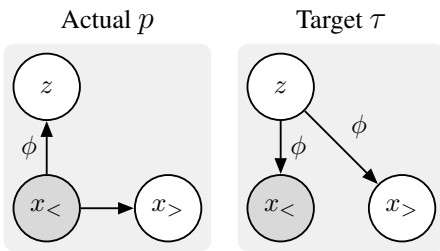

Figure 6: Future Inputs

Figure 6 shows missing inputs on the example of filtering with an HMM (Stratonovich, 1960; Kalman, 1960; Karl et al., 2016), although the same graphical model applies to supervised learning with a BNN

or representation learning with a VAE given train and test data sets. The inputs $x \doteq \{x_<, x_>\}$ consist of past images $x_<$ and future images $x_>$ that follow an unknown and fixed data distribution. We represent the input sequence using a chain $z$ of corresponding compact latent states. However, the representations are computed only based on $x_<$ because $x_>$ is not yet available, as expressed in the factorization of the actual distribution,

$$\text{Actual:} \quad p_\phi(x, z) \doteq \underbrace{p(x_>, x_<)}_{\text{data}} \underbrace{p_\phi(z \mid x_<)}_{\text{belief}},$$

$$\text{Target:} \quad \tau_\phi(x, z) \doteq \underbrace{\tau_\phi(x_< \mid z)}_{\text{likelihood}} \underbrace{\tau_\phi(x_> \mid z)}_{\text{prediction}} \underbrace{\tau(z)}_{\text{prior}}. \tag{14}$$

**Bayesian assumption** Bayesian reasoning operates within the model class $\tau$ and makes the assumption that the model class is correct. Under this assumption, the future inputs $x_> \sim p(x_> \mid x_<, z) = p(x_> \mid x_<)$ follow the target distribution $\tau_\phi(x_> \mid x_<, z) = \tau_\phi(x_> \mid z)$. This renders the divergence of future inputs given the other variables zero, so that $x_>$ does not need to be considered for optimization, recovering standard variational inference from Appendix A.1,

$$\text{KL}\big[p_\phi \;\big\|\; \tau_\phi\big] = \underbrace{\text{KL}\big[p_\phi(x_<, z) \;\big\|\; \tau_\phi(x_<, z)\big]}_{\text{variational inference}} + \underbrace{\text{E} \, \text{KL}\big[p(x_> \mid x_<) \;\big\|\; \tau_\phi(x_> \mid z)\big]}_{\text{uncontrolled future}}. \tag{15}$$

Assuming that future inputs follow the model distribution is appropriate when the model accurately reflects our knowledge about future inputs. However, the assumption does not always hold, for example for data augmentation or distillation (Hinton et al., 2015) that generate data from another distribution to improve the model. Importantly, assuming that future inputs already follow the target is not appropriate when they can be influenced, because there would be no need to intervene.

## A.4 CONTROL

We describe behavior as an optimal control problem where the agent chooses actions to move its distribution of sensory inputs toward a preference distribution over inputs that can be specified via rewards (Morgenstern and Von Neumann, 1953; Lee et al., 2019b). We first cover deterministic actions that lead to KL control (Kappen et al., 2009; Todorov, 2008) and input density exploration (Schmidhuber, 1991; Bellemare et al., 2016; Pathak et al., 2017). Figure 7 shows deterministic control with the input

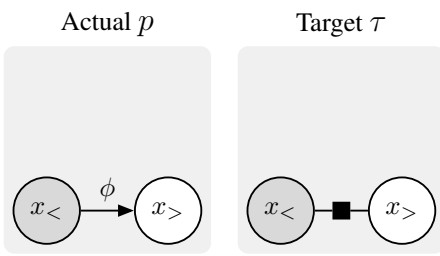

Figure 7: Control

sequence $x \doteq \{x_t\}$ that the agent can partially influence by varying the parameters $\phi$ of the deterministic policy, control rule, or plan. In the graphical model, we group the input sequence into past inputs $x_<$ and future inputs $x_>$. There are no internal latent variables. The target describes the preferences over input sequences that can be unnormalized,

$$\text{Actual:} \quad p_\phi(x) \doteq \prod_t \underbrace{p_\phi(x_t \mid x_{1:t-1})}_{\text{controlled dynamics}},$$

$$\text{Target:} \quad \tau(x) \doteq \prod_t \underbrace{\tau(x_t \mid x_{1:t-1})}_{\text{preferences}}. \tag{16}$$

Minimizing the KL between the actual and target joints maximizes log preferences and the input entropy. Maximizing the input entropy is a simple form of exploration known as input density exploration that encourages rare inputs and aims for a uniform distribution over inputs (Schmidhuber, 1991; Oudeyer et al., 2007). This differs from the action entropy of maximum entropy RL in Section 3.1 and information gain in Appendix A.7 that takes inherent stochasticity into account,

$$\text{KL}\big[p_\phi \;\big\|\; \tau\big] = -\sum_t \Big( \underbrace{\text{E}\big[\ln \tau(x_t \mid x_{1:t-1})\big]}_{\text{expected preferences}} + \underbrace{\text{H}\big[p_\phi(x_t \mid x_{1:t-1})\big]}_{\text{curiosity}} \Big). \tag{17}$$

**Task reward** Motivated by risk-sensitivity (Pratt, 1964; Howard and Matheson, 1972), KL control (Kappen et al., 2009) defines the preferences as exponential task rewards $\tau(x_t \mid x_{1:t-1}) \dot\propto \exp(r(x_t))$. KL-regularized control (Todorov, 2008) defines the preferences with an additional passive dynamics term $\tau(x_t \mid x_{1:t-1}) \dot\propto \exp(r(x_t))\tau'(x_t \mid x_{1:t-1})$. Expected reward (Sutton and Barto, 2018) corresponds to the preferences $\tau_\phi(x_t \mid x_{1:t-1}) \dot\propto \exp(r(x_t))p_\phi(x_t \mid x_{1:t-1})$ that include the controlled dynamics. This cancels out the curiosity term in the joint KL, leading to a simpler objective that does not encourage rare inputs, which might limit exploration of the environment.

**Input density exploration** Under divergence minimization, maximizing the input entropy is not an exploration heuristic but an inherent part of the control objective. In practice, the input entropy is often estimated by learning a density model of individual inputs as in pseudo-counts (Bellemare et al., 2016), latent variable models as in SkewFit (Pong et al., 2019), unnormalized models as in RND (Burda et al., 2018), and non-parameteric models as in reachability (Savinov et al., 2018). More accurately, it can be estimated by a sequence model of inputs as in ICM (Pathak et al., 2017). The expectation over inputs is estimated by sampling episodes from either the actual environment, a replay buffer, or a learned model of the environment (Sutton, 1991).

## A.5 Empowerment

Remaining in the stochastic control setting of Section 3.1, we consider a different target distribution that predicts actions from inputs. This corresponds to an exploration objective that we term generalized empowerment, which maximizes the mutual information between the sequence of future inputs and future actions. It encourages the agent to influence its environment in as many ways as possible while avoiding actions that have no predictable effect.

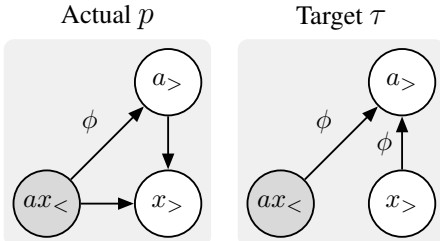

Figure 8: Empowerment

Figure 8 shows stochastic control with an expressive target that captures correlations between inputs and actions. The input sequence is $x \doteq \{x_t\}$ and the action sequence is $a \doteq \{a_t\}$. In the graphical model, these are grouped into past actions and inputs $ax_<$, future actions $a_>$, and future inputs $x_>$. The actual distribution consists of the environment and the stochastic policy. The target predicts actions from the inputs before and after them using a reverse predictor. We use uniform input preferences here, but the target can also include an additional reward factor as in Section 3.1,

$$\text{Actual:} \quad p_\phi(x, a) \doteq \prod_t \underbrace{p(x_t \mid x_{1:t-1}, a_{1:t-1})}_{\text{environment}} \underbrace{p_\phi(a_t \mid x_{1:t}, a_{1:t-1})}_{\text{policy}},$$

$$\text{Target:} \quad \tau_\phi(x, a) \dot\propto \prod_t \underbrace{\tau_\phi(a_t \mid x_{1:T}, a_{1:t-1})}_{\text{reverse predictor}}. \tag{18}$$

Minimizing the joint KL reveals an information boudn between future actions and inputs and a control term that maximizes input entropy and, if specified, task rewards. Empowerment (Klyubin et al., 2005) was originally introduced as potential empowerment to "keep your options open" and was later studied as realized empowerment to "use your options" (Salge et al., 2014). Realized empowerment maximizes the mutual information $\mathrm{I}\big[x_{t+k}; a_{t:t+k} \mid x_{1:t}, a_{1:t-1}\big]$. Divergence minimization generalizes this to the mutual information $\mathrm{I}\big[x_{t:T}; a_{t:T} \mid x_{1:t}, a_{1:t-1}\big]$ between the sequences of future actions and future inputs. The $k$-step variant is recovered by a target that conditions the reverse predictor on fewer inputs. Realized empowerment measures agent's influence on its environment and can be interpreted as maximizing information throughput with the action marginal $p_\phi(a_t \mid a_{t-1})$ as source, the environment as noisy channel, and the reverse predictor as decoder,

$$\mathrm{KL}\big[p_\phi \,\|\, \tau_\phi\big] = \underbrace{\mathrm{E}\,\mathrm{KL}\big[p(x \mid a) \,\|\, \tau(x)\big]}_{\text{control}} - \underbrace{\mathrm{E}\big[\ln \tau_\phi(a \mid x) - \ln p_\phi(a)\big]}_{\text{generalized empowerment}},$$

$$\underbrace{\mathrm{E}\big[\ln \tau_\phi(a \mid x) - \ln p_\phi(a)\big]}_{\text{generalized empowerment}} \geq \sum_t \mathrm{E}\big[\underbrace{\ln \tau_\phi(a_t \mid x, a_{1:t-1})}_{\text{decoder}} - \underbrace{\ln p_\phi(a_t \mid a_{1:t-1})}_{\text{source}}\big]. \tag{19}$$

Empowerment has been studied for continuous state spaces (Salge et al., 2013), for image inputs (Mohamed and Rezende, 2015), optimized using a variational bound (Karl et al., 2017), combined with input density exploration (de Abril and Kanai, 2018) and task rewards (Leibfried et al., 2019), and used for task-agnostic exploration of locomotion behaviors (Zhao et al., 2020). Divergence minimization suggests generalizing empowerment from the input $k$ steps ahead to the sequence of all future inputs. This can be seen as combining empowerment terms of different horizons. Moreover, we offer a principled motivation for combining empowerment with input density exploration. In comparison to maximum entropy RL in Section 3.1, empowerment captures correlations between $x$ and $a$ in its target distribution and thus leads to information maximization. Moreover, it encourages the agent to converge to the input distribution that is proportional to the exponentiated reward.

## A.6 Skill Discovery

Many complex tasks can be broken down into sequences of simpler steps. To leverage this idea, we can condition a policy on temporally abstract options or skills (Sutton et al., 1999). Skill discovery aims to learn useful skills, either for a specific task or without rewards to solve downstream tasks later on. Where empowerment maximizes the mutual information between inputs and actions, skill discovery can be formulated as maximizing the mutual information between inputs and skills (Gregor et al., 2016).

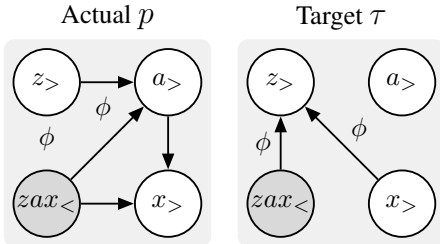

Figure 9: Skill Discovery

Figure 9 shows skill discovery with the input sequence $x \doteq \{x_t\}$, action sequence $a \doteq \{a_t\}$, and the sequence of temporally abstract skills $z \doteq \{z_k\}$. The graphical model groups the sequences into past and future variables. The actual distribution consists of the fixed environment, an abstract policy that selects skills by sampling from a fixed distribution as shown here or as a function of past inputs, and the low-level policy that selects actions based on past inputs and the current skill. The target consists of an action prior and a reverse predictor for the skills and could further include a reward factor,

$$\text{Actual:} \quad p_\phi(x, a, z) \doteq \prod_{k=1}^{T/K} \underbrace{p_\phi(z_k)}_{\text{abstract policy}} \prod_{t=1}^{T} \underbrace{p_\phi(a_t \mid x_{1:t}, a_{1:t-1}, z_{\lfloor t/K \rfloor})}_{\text{policy}} \underbrace{p(x_t \mid x_{1:t-1}, a_{1:t-1})}_{\text{environment}},$$

$$\text{Target:} \quad \tau_\phi(x, a, z) \dot{\propto} \prod_{k=1}^{T/K} \underbrace{\tau_\phi(z_k \mid x)}_{\text{reverse predictor}} \prod_{t=1}^{T} \underbrace{\tau(a_t)}_{\text{action prior}}. \tag{20}$$

Minimizing the joint KL results in a control term as in Appendix A.5, a complexity regularizer for actions as in Section 3.1, and a variational bound on the mutual information between the sequences of inputs and skills. The information bound is a generalization of skill discovery (Gregor et al., 2016; Florensa et al., 2017). Conditioning the reverse predictor only on inputs that align with the duration of the skill recovers skill discovery. Maximizing the mutual information between skills and inputs encourages the agent to learn skills that together realize as many different input sequences as possible while avoiding overlap between the sequences realized by different skills,

$$\text{KL}\big[p_\phi \,\|\, \tau_\phi\big] = \underbrace{\text{E}\,\text{KL}\big[p(x \mid a) \,\|\, \tau(x)\big]}_{\text{control}} + \underbrace{\text{E}\,\text{KL}\big[p_\phi(a \mid x, z) \,\|\, \tau(a)\big]}_{\text{complexity}} - \underbrace{\text{E}\big[\ln \tau_\phi(z \mid x) - \ln p_\phi(z)\big]}_{\text{skill discovery}}. \tag{21}$$

VIC (Gregor et al., 2016) introduced information-based skill discovery as an extension of empowerment, motivating a line of work including SNN (Florensa et al., 2017), DIAYN (Eysenbach et al., 2018), work by Hausman et al. (2018), VALOR (Achiam et al., 2018), and work by Tirumala et al. (2019) and (Shankar and Gupta, 2020). DADS (Sharma et al., 2019) estimates the mutual information in input space by combining a forward predictor of skills with a contrastive bound. Divergence minimization suggests a generalization of skill discovery where actions should not just consider the current skill but also seek out regions of the environment where many skills are applicable.

## A.7 INFORMATION GAIN

Agents need to explore initially unknown environments to achieve goals. Learning about the world is beneficial even when it does not serve maximizing the currently known reward signal, because the knowledge might become useful later on during this or later tasks. Reducing uncertainty requires representing uncertainty about aspects we want to explore, such as dynamics parameters, policy parameters, or state representations. To efficiently reduce uncertainty, the agent should select actions that maximize the expected information gain (Lindley et al., 1956).

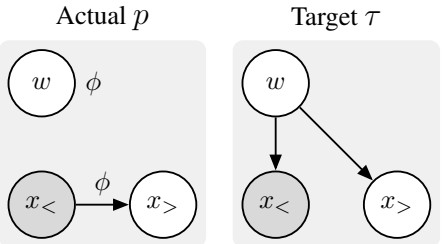

Figure 10: Information Gain

Figure 10 shows information gain exploration on the example of latent model parameters and deterministic actions. The inputs are a sequence $x \doteq \{x_t\}$ and the latent parameters are a global representation $w$. The graphical model separates inputs into past inputs $x_<$ and future inputs $x_>$. The actual distribution consists of the controlled dynamics and the parameter belief. Amortized latent state representations would include a link from $x_<$ to $z$. Latent policy parameters would include a

link from $w$ to $x_>$. The target distribution is a latent variable model that explains past inputs and predicts future inputs, as in Appendix A.3. The target could further include a reward factor,

$$
\begin{aligned}
\text{Actual:} \quad & p_\phi(x, w) \doteq \underbrace{p_\phi(w)}_{\text{belief}} \prod_t \underbrace{p_\phi(x_t \mid x_{1:t-1})}_{\text{controlled dynamics}}, \\
\text{Target:} \quad & \tau(x, w) \doteq \underbrace{\tau(w)}_{\text{prior}} \prod_t \underbrace{\tau(x_t \mid x_{1:t-1}, w)}_{\text{likelihood}}.
\end{aligned}
\tag{22}
$$

Minimizing the KL between the two joints reveals a control term as in previous sections and the information bound between inputs and the latent representation, as derived in Section 2.2. In contrast to Appendix A.3, we can now influence future inputs. This leads to learning representations that are informative of past inputs and exploring future inputs that are informative of the representations. The mutual information between the representation and future inputs is the expected information gain (Lindley et al., 1956; MacKay, 1992b) that encourages inputs that are expected to convey the most bits about the representation to maximally reduce uncertainty in the belief,

$$
\begin{aligned}
\mathrm{KL}\big[p_\phi \,\|\, \tau_\phi\big] \leq & \underbrace{\mathrm{E}\,\mathrm{KL}\big[p_\phi(w \mid x_<) \,\|\, \tau(w)\big]}_{\text{simplicity}} - \underbrace{\mathrm{E}\big[\ln \tau_\phi(x_< \mid w) - \ln p_\phi(x_<)\big]}_{\text{representation learning}} \\
& + \underbrace{\mathrm{E}\,\mathrm{KL}\big[p_\phi(x_> \mid x_<, w) \,\|\, \tau_\phi(x_> \mid x_<)\big]}_{\text{control}} - \underbrace{\mathrm{E}\big[\ln \tau_\phi(w \mid x) - \ln p_\phi(w \mid x_<)\big]}_{\text{information gain}},
\end{aligned}
$$

$$
\underbrace{\mathrm{E}\big[\ln \tau_\phi(w \mid x) - \ln p_\phi(w \mid x_<)\big]}_{\text{information gain}} \geq \sum_{t'>t} \underbrace{\mathrm{E}\big[\ln \tau_\phi(w \mid x_{1:t'}) - \ln p_\phi(w \mid x_{1:t'-1})\big]}_{\text{intrinsic reward}}. \tag{23}
$$

Information gain can be estimated by planning (Sun et al., 2011) or from past environment interaction (Schmidhuber, 1991). State representations lead to agents that disambiguate unobserved environment states, for example by opening doors to see objects behind them, such as in active inference (Da Costa et al., 2020), INDIGO (Azar et al., 2019), and DVBF-LM (Mirchev et al., 2018). Model parameters lead to agents that discover the rules of their environment, such as in active inference (Friston et al., 2015), VIME (Houthooft et al., 2016), MAX (Shyam et al., 2018), and Plan2Explore (Sekar et al., 2020). SLAM resolves uncertainty over both states and dynamics (Moutarlier and Chatila, 1989). Policy parameters lead to agents that explore to find the best behavior, such as bootstrapped DQN (Osband et al., 2016) and Bayesian DQN (Azizzadenesheli et al., 2018).

One might think exploration should seek inputs with large error, but reconstruction and exploration optimize the same objective. Maximizing information gain minimizes the reconstruction error at future time steps by steering toward diverse but predictable inputs. Divergence minimization shows that every latent representation should be accompanied with an expected information gain term, so that the agent optimizes a consistent objective for past and future time steps. Moreover, it shows that representations should be optimized jointly with the policy to support both reconstruction and action choice (Lange and Riedmiller, 2010; Jaderberg et al., 2016; Lee et al., 2019a; Yarats et al., 2019).

## B  ACTIVE INFERENCE

Divergence minimization is motivated by the free energy principle (Friston, 2010; 2019) and its implementation active inference (Friston et al., 2017). Both approaches share the interpretation of models as preferences (Wald, 1947; Brown, 1981; Friston et al., 2012) and account for a variety of intrinsic objectives (Friston et al., 2020). However, typical implementations of active inference have been limited to simple tasks as of today, a problem that divergence minimization overcomes. Active inference differs from divergence minimization in the three aspects discussed below.

**Maximizing the input probability**  Divergence minimization aims to *match* the distribution of the system to the target distribution. Therefore, the agent aims to receive inputs that follow the marginal distribution of inputs under the model. In contrast, active inference aims to *maximize* the probability of inputs under the model. This is often described as minimizing Bayesian surprise. Therefore, the agent aims to receive inputs that are the most probable under its model. Mathematically, this difference stems from the conditional input entropy of the actual system that distinguishes the joint KL divergence in Equation 2 from the expected free energy used in active inference,

$$
\underbrace{\mathrm{KL}\big[p_\phi(x, z) \,\|\, \tau(x, z)\big]}_{\text{joint divergence}} = \underbrace{\mathrm{E}\big[-\ln \tau(x \mid z)\big] + \mathrm{E}\,\mathrm{KL}\big[p_\phi(z \mid x) \,\|\, \tau(z)\big]}_{\text{expected free energy}} - \underbrace{\mathrm{E}\big[-\ln p_\phi(x)\big]}_{\text{input entropy}}. \tag{24}
$$

Both formulations include the entropy of latent variables and thus the information gain that encourages the agent to explore informative future inputs. Moreover, in complex environments, it is unlikely

that the agent ever learns everything so that its beliefs concentrate and it stops exploring. However, in this hypothetical scenario, active inference converges to the input that is most probable under its model. In contrast, divergence minimization aims to converge to sampling from the marginal input distribution under the model, resulting in a larger niche. That said, it is possible to construct a target distribution that includes the input entropy of the actual system and thus overcome this difference.

**Expected free energy action prior**   Divergence minimization optimizes the same objective with respect to representations and actions. Therefore, actions optimize the expected information gain and representations optimize not just past accuracy but also change to support actions in maximizing the expected information gain. In contrast, active inference first optimizes the expected free energy to compute a prior over policies. After that, it optimizes the free energy with respect to both representations and actions. This means active inference optimizes the information gain only with respect to actions, without the representations changing to support better action choice based on future objective terms.

**Bayesian model average over policies**   Typical implementations of active inference compute the action prior using a Bayesian model average. This involves computing the expected free energy for every possible policy or action sequence that is available to the agent. The action prior is then computed as the softmax over the computed values. Enumerating all policies is intractable for larger action spaces or longer planning horizons, thus limiting the applicability of active inference implementations. In contrast, divergence minimization absorbs the objective terms for action and perception into a single variational optimization thereby finessing the computational complexity of computing a separate action prior. This leads to a simple framework, allowing us to draw close connections to the deep RL literature and to scale to challenging tasks, as evidenced by the many established methods that are explained under the divergence minimization framework.

## C   KL INTERPRETATION

Minimizing the KL divergence has a variety of interpretations. In simple terms, it says "optimize a function but don't be too confident." Decomposing Equation 2 shows that we maximize the expected log target while encouraging high entropy of all the random variables. Both terms are expectations under $p_\phi$ and thus depend on the parameter vector $\phi$,

$$\mathrm{KL}\big[p_\phi(x,z) \,\big\|\, \tau(x,z)\big] = \underbrace{\mathrm{E}\big[-\ln\tau(x,z)\big]}_{\text{energy}} - \underbrace{\mathrm{H}\big[x,z\big]}_{\text{entropy}}. \tag{25}$$

The energy term expresses which system configurations we prefer. It is also known as the cross entropy loss, expected log loss, (Bishop, 2006; Murphy, 2012), energy function when unnormalized (LeCun et al., 2006), and agent preferences in control (Morgenstern and Von Neumann, 1953).

The entropy term prevents all random variables in the system from becoming deterministic, encouraging a global search over their possible values. It implements the maximum entropy principle to avoid overconfidence (Jaynes, 1957), Occam's razor to prevent overfitting (Jefferys and Berger, 1992), bounded rationality to halt optimization before reaching the point solution (Ortega and Braun, 2011), and risk-sensitivity to account for model misspecification (Pratt, 1964; Howard and Matheson, 1972).

**Expected utility**   The entropy distinguishes the KL from the expected utility objective that is typical in RL (Sutton and Barto, 2018). Using a distribution as the optimization target is more general, as every system has a distribution but not every system has a utility function it is optimal for. Moreover, the dynamics of any stochastic system maximize only its log stationary distribution (Ao et al., 2013; Friston, 2013; Ma et al., 2015). This motivates using the desired distribution as the optimization target. Expected utility is recovered in the limit of a sharp target that outweighs the entropy.

## D   BACKGROUND

This section introduces notation, defines basic information-theoretic quantities, and briefly reviews KL control and variational inference for latent variable models.

**Expectation**   A random variable $x$ represents an unknown variable that could take on one of multiple values $\bar{x}$, each with an associated probability mass or density $p(x = \bar{x})$. Applying a function to a random variable yields a new random variable $y = f(x)$. The expectation of a random variable is the weighted average of the values it could take on, weighted by their probability,

$$\mathrm{E}\big[f(x)\big] \doteq \int f(x)p(x)\,dx. \tag{26}$$

We use integrals here, as used for random variables that take on continuous values. For discrete variables, the integrals simplify to sums.

**Information** The information of an event $\bar{x}$ measures the number of bits it contains (Shannon, 1948). Intuitively, rare events contain more information. The information is defined as the code length of the event under an optimal encoding for $x \sim p(x)$,

$$\mathrm{I}(\bar{x}) \doteq \ln\left(\frac{1}{p(\bar{x})}\right) = -\ln p(\bar{x}). \tag{27}$$

The logarithm base 2 measures information in bits and the natural base in the unit nats.

**Entropy** The entropy of a random variable $x$ is the expected information of its events. It quantifies the randomness or uncertainty of the random variable. Similarly, the conditional entropy measures the uncertainty of $x$ that we expect to remain after observing another variable $y$,

$$\mathrm{H}\big[x\big] \doteq \mathrm{E}\big[-\ln p(x)\big], \qquad \mathrm{H}\big[x \mid y\big] \doteq \mathrm{E}\big[-\ln p(x \mid y)\big]. \tag{28}$$

Note that the conditional entropy uses an expectation over both variables. A deterministic distribution reaches the minimum entropy of zero. The uniform distribution reaches the maximum entropy, the logarithm of the number of possible events.

**KL divergence** The Kullback-Leibler divergence (Kullback and Leibler, 1951), measures the directed similarity of one distribution to another distribution. The KL divergence is defined as the expectation under $p$ of the log difference between the two distributions $p$ and $\tau$,

$$\mathrm{KL}\big[p(x) \;\big\|\; \tau(x)\big] \doteq \mathrm{E}\big[\ln p(x) - \ln \tau(x)\big] = \mathrm{E}\big[-\ln \tau(x)\big] - \mathrm{H}\big[x\big]. \tag{29}$$

The KL divergence is non-negative and reaches zero if and only if $p = \tau$. Also known as relative entropy, it is the expected number of additional bits needed to describe $x$ when using the code for a different distribution $\tau$ to encode events from $x \sim p(x)$. This is shown by the decomposition as cross-entropy minus entropy shown above. Analogously to the conditional entropy, the conditional KL divergence is an expectation over both variables under the first distribution.

**Mutual information** The mutual information, or simply information, between two random variables $x$ and $y$ measures how many bits the value of $x$ carries about the unobserved value of $y$. It is defined as the entropy of one variable minus its conditional entropy given the other variable,

$$\mathrm{I}\big[X;Y\big] \doteq \mathrm{H}\big[X\big] - \mathrm{H}\big[X \mid Y\big] = \mathrm{E}\big[\ln p(x \mid y) - \ln p(x)\big] = \mathrm{KL}\big[p(x,y) \;\big\|\; p(x)p(y)\big]. \tag{30}$$

The mutual information is symmetric in its arguments and non-negative. It reaches zero if and only if $x$ and $y$ are independent so that $p(x,y) = p(x)p(y)$. Intuitively, it is higher the better we can predict one variable from the other and the more random the variable is by itself. It can also be written as KL divergence between the joint and product of marginals.

**Variational bound** Computing the exact mutual information requires access to both the conditional and marginal distributions. When the conditional is unknown, replacing it with another distribution bounds the mutual information from below (Barber and Agakov, 2003; Poole et al., 2019),

$$\mathrm{I}\big[x;z\big] \geq \mathrm{I}\big[x;z\big] - \mathrm{E}\,\mathrm{KL}\big[p(x \mid z) \;\big\|\; \tau_\phi(x \mid z)\big] = \mathrm{E}\big[\ln \tau_\phi(x \mid z) - \ln p(x)\big]. \tag{31}$$

Maximizing the bound with respect to the parameters $\phi$ tightens the bound, thus bringing $\tau_\phi(x \mid z)$ closer to $p(x \mid z)$. Improving the bound through optimization gives it the name variational bound. The more flexible the family of $\tau_\phi(x \mid z)$, the more accurate the bound can become.

**Dirac distribution** The Dirac distribution (Dirac, 1958), also known as point mass, represents a random variable $x$ with certain event $\bar{x}$. We show an intuitive definition here; for a rigorous definition using measure theory see Rudin (1966),

$$\delta_{\bar{x}}(x) \doteq \begin{cases} 1 & \text{if } x = \bar{x} \\ 0 & \text{else.} \end{cases} \tag{32}$$

The expectation under a Dirac distribution is simply the inner expression evaluated at the certain event, $\mathrm{E}_{\delta_{\bar{x}}(x)}\big[f(x)\big] = f(\bar{x})$. The entropy of a Dirac distributed random variable is therefore $\mathrm{H}\big[x\big] = -\ln \delta_{\bar{x}}(\bar{x}) = 0$ and its mutual information with another random variables is also zero.

**KL control** KL control (Todorov, 2008; Kappen et al., 2009) minimizes the KL divergence between the trajectory $x \sim p_\phi(x)$ of inputs $x \doteq \{x_1, x_2, \ldots, x_T\}$ and a target distribution $\tau(x) \propto \exp(r(x))$ defined with a reward $r(x)$,

$$\mathrm{KL}\big[\underbrace{p_\phi(x)}_{\text{trajectory}} \;\big\|\; \underbrace{\tau(x)}_{\text{target}}\big] = -\underbrace{\mathrm{E}\big[\ln \tau(x)\big]}_{\text{expected reward}} - \underbrace{\mathrm{H}\big[x\big]}_{\text{entropy}}. \tag{33}$$

The KL between the two distributions is minimized with respect to the control rule or action sequence $\phi$, revealing the expected reward and an entropy regularizer. Because the expectations are terms of the trajectory $x$, they are integrals under its distribution $p_\phi$.

**Variational inference** Latent variable models explain inputs $x$ using latent variables $z$. They define a prior $\tau(z)$ and an observation model $\tau(x \mid z)$. To infer the posterior $\tau(z \mid \bar{x})$ that represents a given input $\bar{x}$, we need to condition the model on the input. However, this requires inverting the observation model using Bayes rule and has no closed form in general. To overcome this intractability, variational inference (Hinton and Van Camp, 1993; Jordan et al., 1999) optimizes a parameterized belief $p_\phi(z \mid \bar{x})$ to approximate the posterior by minimizing the KL,

$$\mathrm{KL}\big[p_\phi(z \mid \bar{x}) \,\big\|\, \tau(z \mid \bar{x})\big] + \underbrace{\ln \tau(\bar{x})}_{\text{constant}} = \underbrace{\mathrm{KL}\big[p_\phi(z \mid \bar{x}) \,\big\|\, \tau(z)\big]}_{\text{complexity}} - \underbrace{\mathrm{E}\big[\ln \tau(\bar{x} \mid z)\big]}_{\text{accuracy}}. \tag{34}$$

Adding the marginal $\tau(x)$ that does not depend on $\phi$ completes the intractable posterior to the joint that can be factorized into the available parts $\tau(z)$ and $\tau(x \mid z)$. This reveals a complexity regularizer that keeps the belief close to the prior and an accuracy term that encourages the belief to be representative of the input. This objective is known as the variational free energy or ELBO.

