# OpenReview forum: "Action and Perception as Divergence Minimization"
_ICLR.cc/2021/Conference — Reject_

### Official Review · AnonReviewer2 · 2020-10-27
**Impactful work, but with a few concerns**

**Rating:** 7
**Confidence:** 3

**Review:**

##########################################################################
Summary:

In this manuscript, the authors propose a unifying framework for a large class of inference and reinforcement learning objectives, which have been studied in prior works by various authors. They demonstrate that approaches and central ideas from many different fields in the ML/AI community can be derived as limiting cases of their framework.

##########################################################################
Reasons for score:

Overall, I vote for acceptance (7). Like many, I have employed various variational approaches in the past and see its merit. While I agree with the main idea, this work is not without problems. This is especially problematic for such a broadly applicable work that will most likely influence plenty of future research. My main problems with this submission are:
1.	Presentation. While the paper is, for the most part, well written and well organized, there are some gaps/ jumps that render understanding difficult. Two examples:
A)	The parameters \phi. The authors start by introducing parameters \phi as abstract placeholders for (i) parameters of the true joint distribution of data and latents of the underlying system and (ii) a set of actions an agent can perform to interact with this world. The agent's target distribution has no explicit parameter dependence. So far, so good. Then, one is redirected to the appendix A.1 and A.2. Section A.1 is already a bit confusing because suddenly, additional latents w are introduced that were not mentioned before. Then, suddenly in A.2. the target \tau is suddenly dependent on the parameters \phi, which were initially parameters of the underlying system's true joint distribution. This also happens in Figure 2. C), which is also never referenced in the text. I find this strange mixing of parameters of agent and system very confusing. It also sheds some doubt about the generality of the framework.
B)	I have read the paper carefully and still do not understand Figure 1 completely. This may also be due to the reason that it is only referenced in the appendix. Related: Why does Information gain play such a central role if all derived objectives only contain (upper) bounds for it appear?

2.  	(Unsupported) Claims: In the abstract, the authors promise to offer "a recipe for designing novel objectives". As much as I can see, they only come back to this promise in the conclusion, where they say that one could look at other divergence measures to arrive at new objectives, and they will leave it for future work.
I would not call this a recipe, but an outlook at most.


3.	Too many ideas: It is hard, if not impossible, to explain a broad framework well in a conference proceeding. This work contains so many ideas and establishes many connections that following this work, and understanding them in detail becomes very hard. I would suggest sacrificing some connections in favor of a more concise presentation.
4.	Fixation on KL-divergence: This is more of a suggestion. I understand that many works use the (non-symmetric) KL due to its favorable analytic properties. Thus, I agree that it makes sense to focus this framework on this measure. However, I believe this work's main idea still holds if one would exchange the KL with some other measure of similarity between distributions. Maybe it would make sense to first introduce and discuss the abstract idea of aligning target and belief before fixation on a particular measure. This would also go well with resolving my concern 2..


##########################################################################
Pros:
1.	Unifying framework of many inference, and RL objectives.
2.	Well written.
3.	Will be impactful to a lot of future research.

##########################################################################
Cons:
1.	See my Reasons for score.

##########################################################################
Questions during the rebuttal period:

Please address and clarify the cons above.

#########################################################################

Minor:
· Please consider citing
Toussaint, M., & Storkey, A. (2006). Probabilistic inference for solving discrete and continuous state Markov Decision Processes. International Conference on Machine Learning (ICML), 945–952. https://doi.org/10.1145/1143844.1143963
in the "control as inference" section. To my knowledge, it is one of the first to establish the connection between planning and inference.

---

> ### Author Response · Authors · 2020-11-19
> **Response to AnonReviewer2, Part 2/2**
>
> > In the abstract, the authors promise to offer "a recipe for designing novel objectives". As much as I can see, they only come back to this promise in the conclusion, where they say that one could look at other divergence measures to arrive at new objectives, and they will leave it for future work.
>
> The presented framework suggests at least two ways of deriving novel objectives. The first approach is to study divergence measures other than KL, as you have identified. The second approach is to stick to the KL and change the three components that influence the agent, namely its target distribution, belief family, and optimizer. This is analogous to the specification of a model, belief family, and optimizer in probabilistic modeling and the specification of an architecture, loss function, and optimizer in deep learning. Our paper studies the first component, namely different target distributions. To this end, we motivate the choice of expressive targets (Section 2.2–2.4) and give examples of 8 concrete target distributions (Sections 3 and A.1–7) that result in known objective functions. Due to the page limit, we had to include part of this in the appendix and we are aware that reviewers are not required to read it.
>
> Additionally, when the goal is to design an agent that learns a lot about the world, a practitioner would traditionally look for objective functions in the intrinsic motivation literature. However, it is not clear which of the many proposed objectives will result in an agent that learns the most about the world. Our paper shows that the model by which the agent understands the world is also its target distribution. Thus, the practitioner has a clear path toward implementing the desired agent. This path is to design a powerful world model that after learning assigns high probability to input trajectories in the environment. Using this model as the target distribution defines the agent objective. When additionally trying to solve practical tasks, rewards can be incorporated into the world model as reward factors (e.g. Section 3).
>
> Concrete examples of underexplored world models that we believe are worth implementing in the future include (1) temporally abstract latent state space models that are conditioned on latent skills, (2) hierarchical latent state space model, (3) models that structure their state space into weakly interacting groups that can learn to represent objects, and (3) the use of energy-based models as world models. Minimizing the joint KL to each of these constitutes a novel agent objective with different instances of the representation learning and exploration terms. We have added a paragraph about "Designing novel objectives" in Section 3 to include these intuitions and examples.
>
> > I have read the paper carefully and still do not understand Figure 1 completely. This may also be due to the reason that it is only referenced in the appendix.
>
> Thank you for pointing this out --- we have added a reference to Figure 1 in Section 3. The figure gives an overview of common concrete objectives within our introduced framework. Most of the nodes correspond to a section in the paper (Sections 3 and A.1–7). The two exceptions are "expected reward" (penultimate paragraph of Section A.4) and "maximum likelihood" (last paragraph of Section A.1).

---

> > ### Comment · AnonReviewer2 · 2020-11-24
> > **Follow up response to Part1/2 and Part2/2**
> >
> > Thank you for the clarifications. My concerns have been fully addressed. I stand by my rating.

---

> > > ### Author Response · Authors · 2020-11-24
> > > **Follow-up response to AnonReviewer2**
> > >
> > > Thank you for confirming that we addressed your concerns fully and for confirming your rating!

---

> ### Author Response · Authors · 2020-11-19
> **Response to AnonReviewer2, Part 1/2**
>
> Thank you for your thorough review and insightful suggestions! We have added a paragraph to Section 3 to explain target parameters without having to refer to the appendix and we added a paragraph to summarize the steps for deriving a novel objective in one place. Please let us know if you have further suggestions or questions that we should address.
>
> > in A.2. the target \tau is suddenly dependent on the parameters \phi, which were initially parameters of the underlying system's true joint distribution. [...] I find this strange mixing of parameters of agent and system very confusing. It also sheds some doubt about the generality of the framework.
>
> Thank you for pointing this out as a confusing part in our presentation. We simply mean the idea of a MAP/ML estimate from the probabilistic modeling literature. There are two ways to denote a deterministic parameter estimate. Either use a fixed target distribution with a random variable for which we infer a point mass belief (that has zero entropy), or we parameterize the target using a deterministic parameter. The two are equivalent because in both cases the target receives a deterministic value that has no entropy regularizer. We have added a paragraph on parameterized targets to Section 2 to clarify this in one place, without having to direct readers to the appendix.
>
> > Why does Information gain play such a central role if all derived objectives only contain (upper) bounds for it appear?
>
> The information bounds are variational lower bounds on the mutual information. Taking Eq 3 as an example, the joint divergence is minimized and thus the information bound is maximized. The information bound uses tau for its conditional and is thus a lower bound on the actual mutual information under p, as derived in Eq 30. As such, maximizing them will increase the actual mutual information in the system.
>
> There is an interesting insight here. Estimating mutual information requires predicting one variable from another. The best an agent in an unknown environment can do is to use a model for this prediction, which results exactly in the information bound.

---

### Official Review · AnonReviewer3 · 2020-10-28
**I reject this paper since the formulation of the information-theoretic (i.e., divergence minimisation-based) view of action and perception is already established and well-known.**

**Rating:** 3
**Confidence:** 4

**Review:**

The authors of this paper propose a unified optimisation objective for (sequential) decision-making (i.e., _action_) and representation learning (i.e., _perception_), built on joint (KL) divergence minimisation. As also mentioned by the authors, this is a concept paper and it includes no empirical study.

In particular, the authors demonstrate how existing ideas and approaches to (sequential) decision-making and representation learning can be expressed as a joint KL minimisation problem between a target and "actual" distribution. Such examples are (a) MaxEnt RL, (b) VI, (c) amortised VI, (d) KL control, (e) skill discovery and (f) empowerment, which are all cases of the KL minimisation between a target and an ``actual'' distributions.

**Concerns**:
1. Although the proposed perspective and language is rich and expressive, I question the novelty of the proposed framework, since the information-theoretic view of decision-making and perception is a rather established and old idea, even the term/idea of perception-action cycle is already defined [1]!
2. The power of latent variables for decision-making and their interpretation is also a known idea [1].

**References**

[1] Tishby, N. and Polani, D., 2011. Information theory of decisions and actions. In Perception-action cycle (pp. 601-636). Springer, New York, NY.

---

> ### Author Response · Authors · 2020-11-19
> **Response to AnonReviewer3, Part 2/2**
>
> > The power of latent variables for decision-making and their interpretation is also a known idea [1].
>
> Tishby & Polani (2011) only consider sensory inputs and actions, but not latent representations, such as latent state representations or latent model parameters that are the focus of our work. The research problem of unifying variational representation learning and control as inference has been posed in the SLAC paper (Lee  et al., 2019). The authors suggest a solution, however, they miss the exploration terms that should complement with representation learning terms to result in an objective that is consistent in time.
>
> Our paper considers 3 types of latent variables for decision-making (Section 2.4), namely future actions, future skills, and latent representations. Out of these, the objective function studied by Tishby & Polani (2011) only includes actions and only in the case of a factorized target distribution. Thus, their approach only includes entropy regularization for inputs and actions, but not variational representation learning, information gain, or empowerment. They neither consider latent representations, which are random variables that are never realized (also see Section A.7), nor skills, which are random variables that condition some number of actions and become realized during environment interaction (also see Section A.6).
>
> As our paper shows, an expressive target distribution leads to maximizing the mutual information between the internal latent variables and the sequence of sensory inputs (Section 2.2). This lets us show the different effects of the 3 types of latents on decision making. First, as past actions/skills are observed, their MI with past inputs is constant and they contribute no past terms. Second, latent representations are never observed and thus increase their MI with past inputs via a reconstruction loss. Third, all types of latents increase their MI with future inputs, known as information gain exploration and empowerment (Section 2.4, also see Section A.6–7).
>
> In summary, our paper studies information maximizing agents, offers a solution to the question of integrating latent representations with control as inference as posed by Lee et al. (2019) to unify various KL objectives in the literature, and substantially expands our understanding of the different types of latent variables for decision-making over Tishby & Polani (2011) and other prior works.
>
> References:
> - Tishby & Polani. Information theory of decisions and actions. 2011.
> - Lee et al. Stochastic latent actor-critic: Deep reinforcement learning with a latent variable model. NeurIPS 2020.

---

> > ### Comment · AnonReviewer3 · 2020-11-22
> > **Response to Authors' Rebuttal**
> >
> > Thank you for your reply. I would like to disagree with a few of your comments, please let me know what you think.
> >
> > > At a high level, their paper studies information processing costs of agents, while our paper studies information maximizing agents with the aim of explaining representation learning and directed exploration from a unified perspective.
> >
> > I would like to note that both of you are **minimizing** a divergence which gives rise to the maximization of a mutual information term.
> >
> > > Specifically, they are concerned (1) with sensory inputs and actions only.
> >
> > In their text, they explicitly talk about POMDPs and latent representations, as depicted in (11) and the corresponding references to this figure. Their experiments are using instead MDPs to demonstrate the power of the framework but it would be equally effective in the POMDP setting with latent states.
> >
> > > In contrast, our paper [...] by using an expressive target.
> >
> > Again, their framework doesn't make any assumptions about the target distribution, it's just their experiments that use factorised targets. Comparing the two papers at the level of the framework contribution, I don't see any difference. Also, I cannot compare the two papers in terms of experiments, since you don't have any.
> >
> > Overall, I do agree with the authors on the unification of some prior works. However, I do question the novelty of this! There have been numerous papers, most of which are cited in this paper, where they start from the "Control as Inference" formulation of RL, then they construct a graphical model and perform variational inference (i.e., divergence minimization). I think this paper would serve as a great survey of these prior works as it refreshes older ideas and frameworks and collects recent papers that use these ideas but I don't believe it introduces anything new.
> >
> > **References**
> >
> > [1] Tishby, N. and Polani, D., 2011. Information theory of decisions and actions. In Perception-action cycle (pp. 601-636). Springer, New York, NY.

---

> > > ### Author Response · Authors · 2020-11-23
> > > **Follow-up response to AnonReviewer3**
> > >
> > > > Thank you for your reply. I would like to disagree with a few of your comments, please let me know what you think.
> > >
> > > We sincerely thank you for your response!
> > >
> > > > I would like to note that both of you are minimizing a divergence which gives rise to the maximization of a mutual information term.
> > >
> > > This is incorrect. If the variables have high mutual information under the target distribution, trying to match the target distribution will result in high mutual information in the actual distribution. But if the variables have low mutual information under the target distribution, trying to match the target distribution will result in low mutual information in the actual distribution.
> > >
> > > Tishby & Polani (2011) only consider the case of a factorized target distribution (the unlabeled equation under their Eq 18), which has the least possible amount of mutual information, namely zero. Because of this, their objective aims to minimize the mutual information in the actual system. It is clear from their main objective (unlabeled equation under Eq 26) that their information quantity is minimized, not maximized.
> > >
> > > > Comparing the two papers at the level of the framework contribution, I don't see any difference.
> > >
> > > Are you referring to the KL divergence in their Eq 18 and our Eq 2 as framework? If so, this could be the root cause of our misunderstanding.
> > >
> > > - We understand our framework as the content of Section 2 "Framework", which includes the subsections "Joint KL Minimization", "Information Bounds", "Models as Preferences", and "Past and Future". All these are part of our framework, not just the idea of a trajectory KL that we share with many prior works.
> > > - Similarly, we think of the framework of Tishby & Polani as the KL divergence over MDP trajectories with a target that is factorized in time and uniform over states, because this is the only case analyzed in their equations and is what they motivate as regularizing information processing costs in their abstract.
> > >
> > > Minimizing a KL divergence over trajectories by itself has been known for longer (e.g. Todorov 2008) and is neither a contribution of Tishby & Polani nor of our paper. Is your criticism that from the time the KL over trajectories has been conceived, all objectives that can be derived from it have been known? We believe this reasoning would be flawed --- one might then also say that all objectives have already been known at the time the idea of a cost function was conceived.
> > >
> > > We understand that this misunderstanding could have been caused by us writing the shorthand "framework of divergence minimization" in two places in the paper, instead of the full name of "the framework of action and perception as divergence minimization (APD)" that we otherwise use in our paper. We have updated these two instances to use the full name of the framework, to help prevent our framework, which contains variationally inferred latent representations and emphasizes expressive targets, from being confused with the more generic and previously known idea of a trajectory KL.
> > >
> > > > Again, their framework doesn't make any assumptions about the target distribution, it's just their experiments that use factorised targets.
> > >
> > > Tishby & Polani write "of particular interest are [priors] where the components of the process are independent and "we assume stationarity with partial consistency, where the state distributions are the same for all times, and the action distributions are consistent with them via the policy which we assume constant over time for all t". Thus, it seems incorrect to say that their framework makes no assumptions about the target distribution. It would be difficult to get any interesting properties out of a generic trajectory KL by itself, without additional assumptions.
> > >
> > > > In their text, they explicitly talk about POMDPs and latent representations, as depicted in (11) and the corresponding references to this figure. Their experiments are using instead MDPs to demonstrate the power of the framework but it would be equally effective in the POMDP setting with latent states.
> > >
> > > The objective of Tishby & Polani is based on their Fig 12 not on Fig 11 ("for the current paper, we finally end up at the following diagram which will form the basis for the rest of the paper"). Fig 12 contains only inputs and actions --- quantities that become realized over time. Because of this, their approach cannot describe latents that are variationally inferred together with actions but _are never realized_, such as the latent state estimates and latent parameters that our paper analyzes. This means that even if they had analyzed an expressive target instead of their factorized target, it could only account for empowerment but not expected information gain.
> > >
> > > We would like to thank you again for engaging in the discussion! Please let us know if we have fully addressed your concerns this time, or if you disagree with some of our responses or have further questions for us.

---

> ### Author Response · Authors · 2020-11-19
> **Response to AnonReviewer3, Part 1/2**
>
> Thank you for your comments. We’d like to emphasize that the similarity between our work and that of Tishby & Polani (2011) is small, as discussed below. Moreover, we show that understanding different types of latent variables is an active research problem. Please let us know if this fully addresses your concerns, or if there are other issues that we should address.
>
> > Although the proposed perspective and language is rich and expressive, I question the novelty of the proposed framework, since the information-theoretic view of decision-making and perception is a rather established and old idea, even the term/idea of perception-action cycle is already defined [1]!
>
> Thank you for highlighting the paper by Tishby & Polani (2011). At a high level, their paper studies information processing costs of agents, while our paper studies information maximizing agents with the aim of explaining representation learning and directed exploration from a unified perspective. Specifically, they are concerned (1) with sensory inputs and actions only and (2) aim to *minimize* the multi-information between variables in the system by using a factorized i.i.d target. In contrast, our paper (1) studies agents that variationally infer internal latent representations alongside their actions and (2) *maximize* the mutual information between their internal variables and sensory inputs by using an expressive target.
>
> Their work can be derived as a special case within our framework, as our Section 3 shows. By using an i.i.d factorized target (their Eq 18 and the following unlabeled equation) consisting of an action prior and uniform prior over sensory inputs (first sentence under their Eq 19), they minimize the multi-information of inputs and actions in the MPD. Moreover, they only study the case of a fully observed world state (their Section 5.3.1, third point of the enumerated list). Therefore, their KL objective does not include latent representations (their Eq 18). This results in a policy that maximizes expected reward while maximizing the entropy of sensory inputs and actions (first paragraph of their Section 7 and their Eq 24).
>
> In contrast, our paper studies agents that base their actions on latent representations. To this end, we introduce a unified objective for joint variational inference of representations and actions (our Eq 6). We study the implications of an expressive target and show that it leads to infomax (Section 2.2) via reconstruction and directed exploration (Section 2.4). This establishes a unifying perspective on these two well-established approaches. We then analyze 8 target factorizations (Sections 3 and A1–7), only one of which recovers the case of Tishby & Polani (2011) that neither uses an expressive target nor any latent representations.

---

### Official Review · AnonReviewer1 · 2020-10-28
**This paper discusses about the general training objective for perception, action and beliefs. The authors formulate a joint KL minimization objective and derived modern methods like variational inference,  control, MaxEnt RL to show the correspondance to their unified framework. This work is a theorectical-based paper without empirical studies.**

**Rating:** 6
**Confidence:** 3

**Review:**

The authors proposed to use the joint KL divergence between the generative joint distribution and the target distribution (containing latent variables which could correspond to latent parts we wanted to model (e.g. beliefs). It was illustrative to discuss decomposing the joint KL into different ways and thus forming information bounds in different scenarios. The decomposition of past and future in Eq.6 also provided a unified perspective for looking at the most currently used objectives.

The examples shown in the paper and appendix give a good illustration of how people can make assumptions or design the terms to convert prevalent objectives into objectives that follow from this joint KL divergence framework. This is, in my mind, one of their key contributions for connecting the past progress in a general and unified way.

However, one concern about this paper is that the proposal of such a unified KL minimization framework is in fact a bit too general and abstract. In fact, many methods mentioned in this work shared a similar insight of deriving objectives from a KL-minimization perspective, but some factors are omitted to better fit the corresponding tasks. The general decomposition discussed in this paper provides little hint on how new objectives could be derived for problems. The general framework does somehow serve as the guideline, but my worry is that its effect will be limited as we still need to design the mapping for the terms in the general objective accordingly in different tasks.

Given the pros and cons of this paper, I'm putting a borderline decision for now. The authors should clear any of my misunderstandings and perhaps show the potential for this general framework as a source for new objectives.

=====================================================================================================

After reading the authors rebuttal, my major concerns are fully addressed and I decide to keep my decision as weak accept

---

> ### Author Response · Authors · 2020-11-19
> **Response to AnonReviewer1, Part 2/2**
>
> > The general decomposition discussed in this paper provides little hint on how new objectives could be derived for problems. The general framework does somehow serve as the guideline, but my worry is that its effect will be limited as we still need to design the mapping for the terms in the general objective accordingly in different tasks.
>
> The implementation of an agent in practice is influenced by three components, namely the agent's target distribution, belief family, and optimizer. This is analogous to the specification of a model, belief family, and optimizer in probabilistic modeling and the specification of an architecture, loss function, and optimizer in deep learning. Our paper studies the first component, namely different target distributions. To this end, we motivate the choice of expressive targets (Section 2.2–2.4) and give examples of 8 concrete target distributions (Sections 3 and A.1–7) that result in known objective functions. Due to the page limit, we had to include part of this in the appendix and we are aware that reviewers are not required to read it.
>
> Additionally, when the goal is to design an agent that learns a lot about the world, a practitioner would traditionally look for objective functions in the intrinsic motivation literature. However, it is not clear which of the many proposed objectives will result in an agent that learns the most about the world. Our paper shows that the model by which the agent understands the world is also its target distribution. Thus, the practitioner has a clear path toward implementing the desired agent. This path is to design a powerful world model that after learning assigns high probability to input trajectories in the environment. Using this model as the target distribution defines the agent objective. When additionally trying to solve practical tasks, rewards can be incorporated into the world model as reward factors (e.g. Section 3).
>
> Concrete examples of under-explored world models that we believe are worth implementing in the future include (1) temporally abstract latent state space models that are conditioned on latent skills, (2) hierarchical latent state space model, (3) models that structure their state space into weakly interacting groups that can learn to represent objects, and (3) the use of energy-based models as world models. Minimizing the joint KL to each of these constitutes a novel agent objective with different instances of the representation learning and exploration terms. We will add a paragraph on "Designing novel objectives" to the paper to include these intuitions and examples.
>
> > Given the pros and cons of this paper, I'm putting a borderline decision for now. The authors should clear any of my misunderstandings and perhaps show the potential for this general framework as a source for new objectives.
>
> We hope that we have cleared up your concern about the relation between the joint objective for representation learning and control we study and previous applications of KL minimization. We hope the examples of novel agent objectives have helped make leveraging the framework concrete. Please let us know if this addresses your concerns or whether there are any other points we should address.
>
> References:
> - Lee et al. Stochastic latent actor-critic: Deep reinforcement learning with a latent variable model. NeurIPS 2020.
> - Houthooft et al. VIME: Variational Information Maximizing Exploration. NeurIPS 2016.
> - Sekar et al. Planning to Explore via Self-Supervised World Models. ICML 2020.

---

> > ### Comment · AnonReviewer1 · 2020-11-24
> > **Response to the author**
> >
> > Thank you for your clarification and further illustration. I agree with your thoughts on having a unified perspective to look at problems when trying to figure out what kind of optimization problem, and I believe treating it as a distribution fitting problem is definitely one way toward most problems. This further leads to the provided examples in the appendix, showing its capability.  I guess when facing different scenarios, we should not hope a general framework is capable of doing it all, instead, it should work as a general heuristic that affects our derivation direction.

---

> > > ### Author Response · Authors · 2020-11-24
> > > **Follow-up response to AnonReviewer1**
> > >
> > > Thank you for your response and for agreeing on the value of the unified perspective presented in this paper! Could you please confirm whether our response fully addressed your concerns or if there are any remaining concerns or questions --- that lead to you continuing to recommend a weak accept --- that we could address?

---

> > > > ### Comment · AnonReviewer1 · 2020-11-24
> > > > **Final comment**
> > > >
> > > > My concerns are fully addressed, thank you. But I do agree with R3 to a certain extent that this work is based on many similar prior works that solved problems with a similar intuition, and could be more impactful as a survey paper (which of course, has its own value). I guess this leads to the recommendation for weak-accept.

---

> > > > > ### Author Response · Authors · 2020-11-24
> > > > > **Final response**
> > > > >
> > > > > Thank you for confirming that we fully addressed your comments, for confirming your recommendation for the paper, and for engaging in the discussion!

---

> ### Author Response · Authors · 2020-11-19
> **Response to AnonReviewer1, Part 1/2**
>
> Thank you for your review! We elaborate below that unifying previous applications of KL minimization for either representation learning and control was an open problem. We also describe ideas for novel objectives that correspond to target distributions within our framework. Please let us know if this fully addresses your concerns or whether there are any other issues we should address.
>
> > The authors proposed to use the joint KL divergence between the generative joint distribution and the target distribution (containing latent variables which could correspond to latent parts we wanted to model (e.g. beliefs). It was illustrative to discuss decomposing the joint KL into different ways and thus forming information bounds in different scenarios. The decomposition of past and future in Eq.6 also provided a unified perspective for looking at the most currently used objectives`.
>
> Thank you for the accurate summary.
>
> > one concern about this paper is that the proposal of such a unified KL minimization framework is in fact a bit too general and abstract. In fact, many methods mentioned in this work shared a similar insight of deriving objectives from a KL-minimization perspective, but some factors are omitted to better fit the corresponding tasks.
>
> We agree that joint KL minimization is a very general starting point. We then add constraints to the problem, such as the restriction to an expressive target (Section 2.2) or specific target factorizations (Sections 3 and A1–7). We further agree that many of the individual objectives that can be derived this way have previously been proposed in the literature for solving specific problems and shown empirical success.
>
> However, we believe that the general perspective presented in our paper has been an open research problem rather than being widely known. Most previous objectives that have been derived from the KL divergence have either been targeting representation learning (e.g. state representations, model parameters) or control (e.g. actions, temporally-abstract skills), but not both.
>
> In fact, Lee et al. (2019) pose unifying inference of representation and control as an open research problem. Our perspective unifies the two in a joint variational inference problem and identifies the future information term that is missing in the work of Lee et al., which leads to directed exploration and empowerment, and ensures a consistent objective that does not change across time steps. These exploration methods have been shown to be empirically successful outside of the KL minimization framework (e.g. Houthooft et al., 2016, Sekar et al. 2020).

---

### Official Review · AnonReviewer4 · 2020-11-03
**Unifying information-theoretic objective**

**Rating:** 6
**Confidence:** 4

**Review:**

The authors formulate a general framework that unifies inference, action/perception, control, and several other tasks. The framework is based on minimizing the KL divergence between a parameterized "actual" distribution and a "target" distribution. The authors argue that this formulation unifies a wide range of previously proposed objectives. They also argue that it has some advantages when compared to Friston's "free energy principle" framework, with which it shares many similarities, in particular that probability matching is preferred to surprise minimization.

The paper is clearly-written and provides a very thorough literature review. However, generally I question the scientific value of such all-encompassing unifying frameworks, and this paper in particular offers no concrete formal or empirical results, while promising a lot. At the end of the day, the divergence minimization objective is nothing more than MaxEnt, decorated with various interpretations and decompositions. Without empirical support, I do not find the interpretations and decompositions very convincing -- as one example, does divergence minimization *really* mean that "expressive world models lead to autonomous agents that understand and inhabit large niches"?

One of the issues is that the paper appears to treat the "heart of the matter" (i.e., the source of interesting solutions) as if it lay in the elegant and generic objective. In my opinion, however, the real heart of the matter will be encoded in (1) the structure of the target distribution, (2) the structure/parameterization of the actual distribution, and (3) the optimization algorithm that can actually minimize the (typically) high-dimensional  objective. The quality of resulting solutions depend on 1-3 -- all of which need to be exogenously specified --- because divergence minimization cannot on its own produce interesting behavior.

At the end of the day, I do think there is some value in providing a unifying framework, and developing information-theoretic decompositions and interpretation. However, I think the paper would be *much* stronger if it was considerably longer and had more room to breathe (which it doesn't have right now -- given all the connections it tries to make), and if qualitative statements (of the type discussed above) were accompanied by empirical results (even if simulations with simple toy models).

---

> ### Author Response · Authors · 2020-11-19
> **Response to AnonReviewer4, Part 2/2**
>
> > the real heart of the matter will be encoded in (1) the structure of the target distribution, (2) the structure/parameterization of the actual distribution, and (3) the optimization algorithm that can actually minimize the (typically) high-dimensional objective
>
> That is absolutely correct --- these three components determine the agent's behavior in practice. We advocate breaking down the problem of designing agents into the three components, as it allows studying the components separately and mix-and-match them when implementing agents. This is analogous to the specification of a model, belief family, and optimizer in probabilistic modeling and the specification of an architecture, loss function, and optimizer in deep learning. We believe that breaking down the problem of designing agents into the structure of the target distribution, parameterization of the actual distribution, and the optimization algorithm, as suggested by the APD framework, can accelerate progress in reinforcement learning research.
>
> Our paper studies the first of the three components you mentioned, namely the target distribution, we summarized in our response above. We completely agree that the remaining two components also matter in practice and are worth studying. Our framework provides a map of the possible agent objectives that result from expressive target distributions to guide future work in empirically exploring the space of possible agent implementations in a structured manner.
>
> > At the end of the day, I do think there is some value in providing a unifying framework, and developing information-theoretic decompositions and interpretation. However, I think the paper would be much stronger if it was considerably longer and had more room to breathe (which it doesn't have right now -- given all the connections it tries to make), and if qualitative statements (of the type discussed above) were accompanied by empirical results (even if simulations with simple toy models).
>
> We completely agree that having more space would allow for more detailed explanations. Unfortunately, we are bound to the page limit of ICLR in this regard. That said, we are allowed an additional page for the final version of the paper. We will leverage this page to include an additional example of a target factorization from the appendix into the main paper.
>
> While working on this paper, we concluded that empirical simulations would likely distract from the main message of the paper. While we could implement concrete instances of APD, such as the target factorizations studied in Sections 3 and A1–7, these objectives have successful implementations in the prior literature already. Instead, we prefer to use the available page limit for developing our main contribution, which is to analyze the effects of various constraints on the target distribution and to develop a unified perspective on many existing algorithms in the literature. We believe these conceptual contributions offer substantial value for the reinforcement learning community and stand on their own.
>
> References:
> - Lee et al. Stochastic latent actor-critic: Deep reinforcement learning with a latent variable model. NeurIPS 2020.
> - Haarnoja et al. Soft actor-critic: Off-policy maximum entropy deep reinforcement learning with a stochastic actor. ICML 2018.

---

> ### Author Response · Authors · 2020-11-19
> **Response to AnonReviewer4, Part 1/2**
>
> We thank you for your review! Your summary of our paper is accurate. We emphasize that despite its general starting point, our paper studies concrete constraints on the target distribution and their implications on the resulting practical objective functions, as detailed below. Please let us know if this fully addresses your concerns or if there are any other points we should address!
>
> > At the end of the day, the divergence minimization objective is nothing more than MaxEnt, decorated with various interpretations and decompositions. Without empirical support, I do not find the interpretations and decompositions very convincing
>
> Our paper uses as a general starting point the joint KL divergence or maximum entropy principle. From this starting point, we impose additional constraints on the target distribution and analyze the resulting objective functions. This is important because the choice of target distribution likely has a larger effect on the resulting agent behavior than the entropy regularizer by itself. The content of our paper therefore goes beyond the maximum entropy principle. Specifically, we study the implications of using an "expressive target distribution" and we analyze 8 concrete factorizations for the target distributions, as summarized below.
>
> 1) In Section 2.2, we show that expressive targets, that capture dependencies between latents and inputs, lead to information maximization. In Eq 6, we show that this accounts for joint representation learning, control, and directed exploration. This answers the problem of unifying representation learning with control as inference as recently posed in the literature by SLAC (Lee et al., 2019), who miss the infogain and empowerment exploration terms that we identify.
>
> 2) To achieve complex goals, an agent needs to absorb information about the environment into its internal state. Our analysis shows that currently common RL algorithms, such as SAC (Haarnoja et al., 2018), correspond to factorized targets and thus do not explicitly aim to increase their mutual information with the environment. Instead, it suggests the importance of expressive targets, such as powerful latent variable models or "world models" of the input sequence as target distributions.
>
> 3) We study 8 different factorizations of the target distribution in Sections 3 and A1–7 by recovering various concrete objectives used in the literature. This exposes the assumptions behind practical algorithms, illustrates the effects of different types of latent variables (actions, skills, representations) for decision making, and empowers practitioners to derive novel objectives by choosing appropriate target factorizations for that problem at hand.
>
> We had to include parts of 3) in the appendix due to the page limit and we know that reviewers are not required to read this. For this reason, we included an intuitive summary at the end of Section 2.4. The final version of the paper will be allowed an additional page, which allows us to include one additional example in the main text, such as Section A.5, A.6 or A.7.
>
> We also acknowledge that many forms of KL minimization have been studied in the literature to formulate objectives for either perception or action, and that these have led to many empirically successful algorithms (Section 2.4, 3, A1–7). This prior work allowed us to focus on the conceptual unification of KL objectives for perception and action in this paper, without the need for empirical experiments. Our paper introduces a unified KL minimization perspective that incorporates both perception and action that has previously been missing from the literature.

---

### Decision · Program_Chairs · 2021-01-07
**Final Decision**

**Decision:**

Reject

**Comment:**

The paper presents an KL-divergence minimisation approach to the action–perception loop, and thus presents a unifying view on concepts such as Empowerment, entropy-based RL, optimal control, etc.  The paper does two things here: it serves as a survey paper, but on top of that puts these in a unifying theory.  While the direct merit of that may not be obvious, it does serve as a good basis to combine the fields more formally.

Unfortunately, the paper suffers from the length restrictions.  With more than half of the paper in the appendix, it should be published at a journal or directly at arXiv.   Not having a page limit would improve the readability much.  ICLR may not be the best venue for review papers.